# An original infection model identifies host lipoprotein import as a route for blood-brain barrier crossing

Billel Benmimoun [1], Florentia Papastefanaki [2], Bruno Périchon [3], Katerina Segklia [2], Nicolas Roby [1], Vivi Miriagou [4], Christine Schmitt[5], Shaynoor Dramsi [3], Rebecca Matsas [2] & Pauline Spéder [1✉]

Pathogens able to cross the blood-brain barrier (BBB) induce long-term neurological sequelae and death. Understanding how neurotropic pathogens bypass this strong physiological barrier is a prerequisite to devise therapeutic strategies. Here we propose an innovative model of infection in the developing *Drosophila* brain, combining whole brain explants with in vivo systemic infection. We find that several mammalian pathogens are able to cross the *Drosophila* BBB, including Group B *Streptococcus* (GBS). Amongst GBS surface components, lipoproteins, and in particular the B leucine-rich Blr, are important for BBB crossing and virulence in *Drosophila*. Further, we identify (V)LDL receptor LpR2, expressed in the BBB, as a host receptor for Blr, allowing GBS translocation through endocytosis. Finally, we show that Blr is required for BBB crossing and pathogenicity in a murine model of infection. Our results demonstrate the potential of *Drosophila* for studying BBB crossing by pathogens and identify a new mechanism by which pathogens exploit the machinery of host barriers to generate brain infection.

[1] Institut Pasteur, Brain Plasticity in Response to the Environment, CNRS, UMR3738 Paris, France. [2] Laboratory of Cellular and Molecular Neurobiology-Stem Cells, Department of Neurobiology, Hellenic Pasteur Institute, Athens, Greece. [3] Unité de Biologie des Bactéries Pathogènes à Gram-positif, Institut Pasteur, CNRS, UMR 2001 Paris, France. [4] Laboratory of Bacteriology, Department of Microbiology, Hellenic Pasteur Institute, Athens, Greece. [5] Ultrastructure UTechS Ultrastructural Bioimaging Platform, Institut Pasteur, Paris, France. ✉email: pauline.speder@pasteur.fr

Central nervous system (CNS) infections are rare, yet extremely damaging. They lead to fatal outcomes and long-term neurological disabilities in surviving infants and adults, including cognitive deficit and motor impairment[1]. A major route for CNS infection is the bloodstream, which pathogens enter after crossing the epithelial barriers of the skin and gut[2,3], and in which they circulate as free particles or carried by blood cells[4]. To infect the brain, pathogens must ultimately bypass an additional guardian: the blood–brain barrier (BBB)[5]. The BBB is both a selective physical and chemical filter controlling molecular import into the CNS, thus enabling neuroprotective functions[6]. In higher vertebrates, brain microvascular endothelial cells form the core structure of the BBB. These cells are equipped to provide selective insulation, harbouring intercellular tight junctions, absence of fenestrae, and asymmetrically localised transport systems[6,7]. The BBB also includes perivascular pericytes, astrocytes and a basal membrane made of the extracellular matrix, which regulate BBB integrity and functions[8]. This complex set of interlinked layers behaves as a double-edge sword for the organism: it restricts the entry of pathogens as well as therapeutic molecules, such as antibiotics[7].

Pathogens that manage to cross the BBB thus secure their access to the CNS, where they tend to be immunologically protected. Accordingly, neuro-invasive, neurotropic pathogens have developed intricate mechanisms that allow them to cross this layer and invade the CNS[2,3,5]. Three main strategies have been proposed so far: transcellular, paracellular and Trojan horse. The transcellular entry occurs through a receptor-mediated mechanism or pinocytosis, while the paracellular mechanism follows the increase of BBB permeability due to tight junction disruption. The Trojan horse mechanism uses infected blood cells which transmigrate from the periphery to the CNS. Pathogens could actually use several of these routes to invade the brain[5].

So far, most of this knowledge comes from in vitro models of BBB[9,10] where a monolayer of endothelial cells is co-cultured with pericytes and astrocytes in transwells[11]. However, they display a lot of variations in their tightness and thus reproducibility is a major issue. Despite induced pluripotent stem cell-related advances[11] and new set-ups like microfluidic organ-on-chips[12], these models struggle to recapitulate complex parameters crucial to BBB properties, including 3D architecture and dynamic cellular interactions. Animal models, mostly mice and rats, but also zebrafish, exist and have provided essential contributions to mechanistic explorations[2,10], including revisiting results from in vitro models[13]. Manipulating these organisms to reach a cellular resolution and causal relationships is nevertheless still highly challenging. Cost and ethical issues also hinder their extensive use.

*Drosophila* is a powerful and tractable model system, with unrivalled genetics. It has been very successful in identifying conserved molecular mechanisms in innate immunity, such as the Toll pathway[14], with a focus on systemic and epithelial immunity (skin and gut). Strikingly, many aspects of mammalian neurogenesis are conserved in the *Drosophila* larva CNS, a post-embryonic, juvenile stage which also harbours a BBB (Fig. 1a, b). The open circulatory system of the fly carries the haemolymph, which is in direct contact with all the organs including the CNS. The BBB represents its outermost structure and is composed of two glial layers[15]. The subperineurial glia (SPG) are large polarised cells forming an epithelium-like structure with septate junctions (Fig. 1c), the equivalent of tight junctions in vertebrates. These represent a physical barrier to paracellular diffusion, similarly to the mammalian brain vascular endothelium[16,17]. The perineurial glia (PG) cover the SPG and are proposed to be a haemolymph sensor[18,19]. Several studies have now uncovered a striking conservation of molecules and import mechanisms

between fly and mouse BBB cells[20,21]. Thus, the *Drosophila* BBB represents a physical and chemical barrier that retains conserved chemoprotective strategies with the mammalian BBB, ensuring brain homoeostasis.

Here we show that the *Drosophila* larval brain is a relevant and valuable system to model brain infection and discover cellular mechanisms of BBB crossing by mammalian pathogens. Taking advantage of this innovative model, we identified the lipoprotein Blr as a new virulence factor contributing to GBS neurotropism in the fly and mouse. We further identified the *Drosophila* lipoprotein receptor LpR2 as a host receptor for Blr in the BBB, mediating GBS internalisation through endocytosis.

## Results

**Group B *Streptococcus* actively invades the *Drosophila* larval brain in an explant set-up.** Establishing a model of brain infection in the fly larva required experimental set-ups in which whole, intact living brains would be in contact with pathogens. We first devised an ex vivo protocol, as a straightforward platform for screening pathogens and conditions (Fig. 1d). Whole third instar larvae were opened posteriorly to expose all tissues while preventing damages to the brain and minimising injuries of the peripheral nerves (see Methods section). These brain explants were transferred to culture conditions that preserve cell viability, cell proliferation and BBB permeability, and that do not induce oxidative stress (Supplementary Fig. 1a–d). The culture medium was then inoculated with the chosen pathogens at selected doses at 30 °C, close to mammalian body temperature yet tolerated by *Drosophila*. Brain explants were left in contact with pathogens for a given time (usually 3 h) to allow binding, washed to remove unattached microorganisms and kept in culture until the desired time of analysis. Whole fixed brains were analysed under confocal microscopy in order to distinguish brain entry from adhesion and precisely localise and quantify individual pathogens (Fig. 1d and see Methods section).

We used this set-up to screen for prokaryotic or eukaryotic pathogens known to trigger encephalitis and/or meningitis in mammals. We found that several were able to cross the *Drosophila* BBB and generate brain infection 24 h after inoculation (*Streptococcus agalactiae*, *Streptococcus pneumoniae*, *Neisseria meningitidis*, *Listeria monocytogenes*, *Candida glabrata* and non-hyphal *Candida albicans*; Fig. 1e). In contrast, non-pathogenic strains (*Lactobacillus plantarum*, non-pathogenic *Escherichia coli*, and *Saccharomyces cerevisiae*) were not able to enter the *Drosophila* brain, pointing to specific entry mechanisms under these conditions.

Amongst the various pathogens tested, *S. agalactiae* (Group B *Streptococcus*, GBS) proved to be the most efficient to cross the fly BBB and was detected both inside the brain and attached to its surface (Fig. 2a). We thus focused on GBS, an opportunistic gram-positive bacterium responsible for severe invasive infections in neonates leading to pneumonia, septicaemia and meningitis[22–24]. Despite available antibiotic treatments and intrapartum prophylaxis, these cases still represent 10% of mortality and neurological sequelae in 25 to 50% of survivors[25,26]. Among the various clinical isolates tested, NEM316, being the most efficient to infect *Drosophila* larval brain explants, was chosen as our reference GBS strain (Fig. 2b, from now on called GBS). Interestingly, dead, formaldehyde-fixed GBS were unable to enter *Drosophila* brain explants, underlining the fact that GBS needs to be alive to cross the BBB.

**Dissecting GBS strategies to cross the multiple layers of the BBB.** Using confocal and specific markers of the extracellular matrix (ECM), PG and SPG layers, as well as transmission

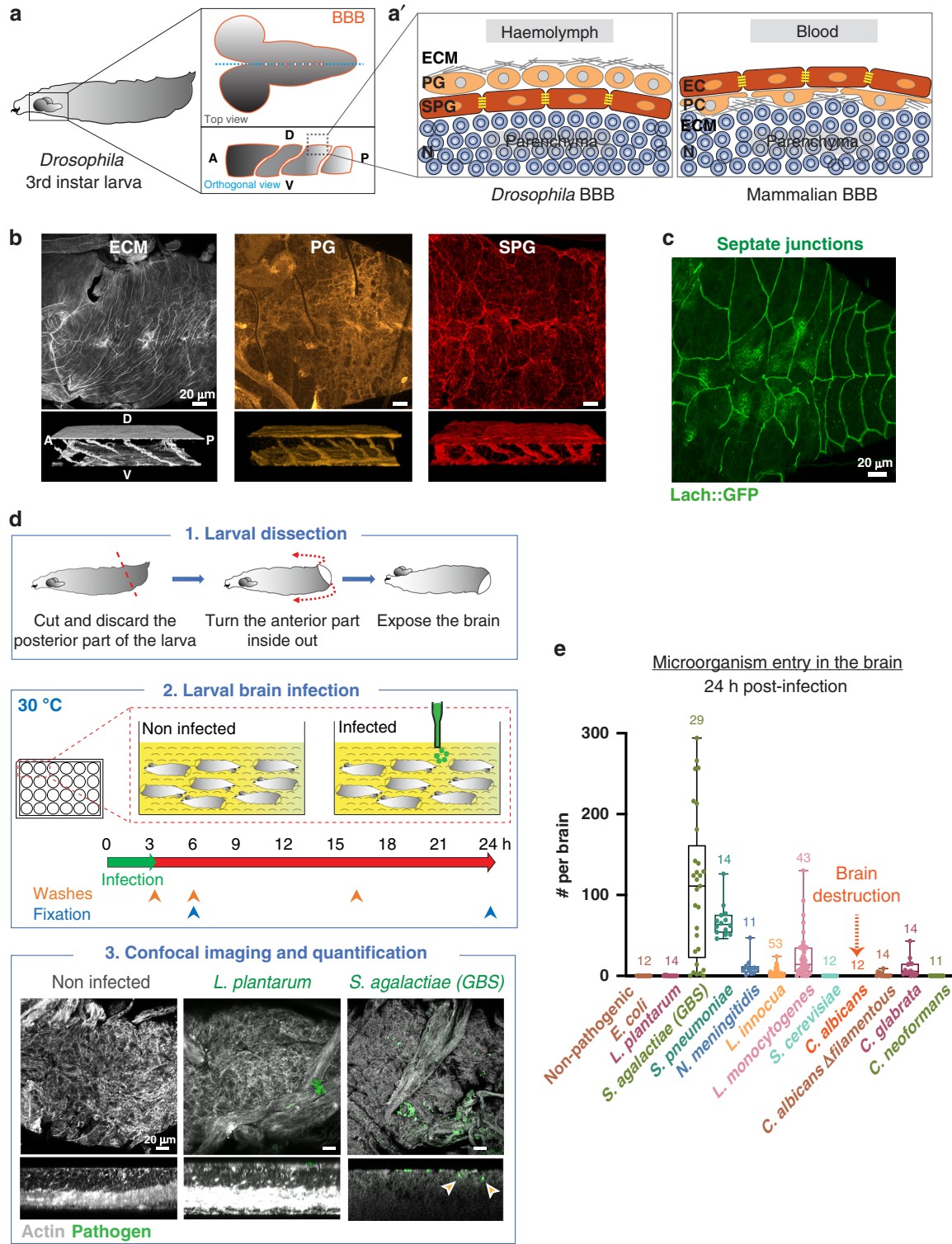

electron microscopy, we noticed several morphological disruptions under GBS infection. First, using protein trap lines (*vkg::GFP*[27] and *trol::GFP*[28]) to visualise conserved components of the ECM (respectively collagen IV and heparan sulfate proteoglycan (HSPG) Perlecan), we revealed: (i) that GBS was laying on or embedded in the ECM (Fig. 2c) and (ii) that both the overall Collagen IV and Perlecan networks were disrupted (Supplementary Fig. 2a, b) and appeared locally clumped around the embedded bacteria (Fig. 2c and Supplementary Fig. 2c). In

addition, we observed a decrease in signal intensity of Trol::GFP (Supplementary Fig. 2b, c). Second, analysis of the cellular layers showed alterations in membrane morphology under GBS infection. Indeed, staining for a PG membrane reporter revealed a partial and fainter signal under infection (Fig. 2d). Furthermore, we noticed in some cases an altered morphology of SPG membrane and septate junctions (Fig. 2e and Supplementary Fig. 2d, e).

These observations led us to investigate GBS transport means across the PG and SPG. The first cellular layer to cross, the PG,

**Fig. 1 Screening for mammalian neuro-invasive pathogens in a brain explant set-up identifies Group B *Streptococcus* as able to cross the *Drosophila* blood–brain barrier. a** Schematic representation of *Drosophila* third instar larva showing the brain suspended in the haemolymph. Top and orthogonal views of the brain covered by the BBB (dark orange). A, anterior. P, posterior. D, dorsal. V, ventral. **a′** Schematic representations of the composite *Drosophila* and mammalian BBBs, which include a layer of extracellular matrix (ECM in grey), a regulatory layer (perineurial glia (PG) and pericytes (PC) in light orange) and a barrier layer (subperineurial glia (SPG) and endothelial cells (EC), dark orange) harbouring strong cell junctions (septate junctions and tight junctions, yellow). Neurons (N), one major type of brain cell populations, are illustrated in blue. **b** Confocal images of the *Drosophila* BBB (top view and 3D orthogonal view) labelled for the ECM in grey (*vkg::GFP*), the PG in light orange (*NP6293-GAL4>mCD8-GFP*) and the SPG in red (*mdr65-mtd-tomato*). **c** Septate junctions in green (*Lachesin::GFP*). **d** Ex vivo protocol. Step 1 depicts the dissection method used to expose the brain while minimising damages. Step 2 illustrates the culture and infection protocols. Step 3 depicts confocal images of the brains (top and orthogonal views) stained with phalloidin (white). Bacteria (*L. plantarum* and *S. agalactiae [GBS]*) are stained in green. Orange arrows show GBS inside the brain. **e** Screening mammalian neurotropic pathogens for their ability to cross the *Drosophila* BBB. *S. agalactiae* (GBS), *S. pneumoniae*, *N. meningitidis*, *L. monocytogenes* and *C. glabrata* were able to cross the *Drosophila* BBB. *C. albicans* exists in a filamentous, hyphal form linked to pathogenicity, which destroyed the brain, and in yeast, non-hyphal form (*C. albicans Δfilamentous*) which entered the brain. *L. innocua* rarely crossed the BBB, while non-pathogenic *E. coli, L. plantarum, S. cerevisiae* and *C. neoformans* were not able to invade the brain. Results are presented as box and whisker plots. Whiskers mark the minimum and maximum, the box includes the 25th–75th percentile, and the line in the box is the median. Numbers above the boxes represent the number of larvae analysed. Source data are provided as a Source Data file for **e**.

does not have intercellular junctions and thus a paracellular route could be used. To determine whether GBS relies on internalisation to cross the PG, we blocked endocytosis specifically in this layer by preventing dynamin (*Drosophila* Shibire) function through the overexpression of its temperature-sensitive non-functional form[29] (*shibire*[ts]). We found that preventing endocytosis in the PG did not alter brain entry (Fig. 2f), suggesting that indeed GBS does not rely on an intracellular route in this layer. Ultimately, the capacity of GBS to invade the brain and generate infection is linked to its ability to cross the SPG, e.g. the BBB physical barrier per se[2,24]. Thus, we next assayed BBB permeability of brain explants by dextran diffusion, and found a significant increase during GBS infection (Fig. 2g), suggesting that GBS affects the uptake of molecules across the BBB in general.

Intriguingly, we noticed that the morphological alterations were not particularly associated with GBS localisation, but occurred brain-wide, suggesting a systemic origin. Extracellular acidosis has been shown to build in the brain under meningitis[30,31], and GBS is known to secrete lactic acid and acidify the culture environment[26]. Therefore, we measured the pH of the culture medium 3 h post-infection and revealed strong acidification (Supplementary Fig. 2f). Blocking medium acidification using a HEPES buffer rescued Collagen IV general pattern (Supplementary Fig. 2a), however, localised clumping still remained around GBS (Fig. 2c). These results suggest that GBS could locally alter the ECM to facilitate access to the cellular layers. Strikingly, medium buffering considerably restored PG structure as well as SPG membrane and septate junction morphology (Fig. 2d, e and Supplementary Fig. 2d, e). Consistent with these observations, blocking medium acidification strongly prevented the increase in BBB permeability observed under GBS infection (Fig. 2g). Interestingly, bacterial counts in the brain also significantly changed in the buffered medium compared to non-buffered (Fig. 2h). However, this change was moderate, and GBS still successfully entered the brain. Altogether these data show that, although acidification might help GBS by altering BBB features, it is not a prerequisite for brain entry. This argues for the critical involvement of specific mechanisms for SPG crossing by GBS on top of acidity-induced host tissue alterations, including ECM rearrangement and increase in BBB permeability.

**The B Streptococcal surface lipoprotein Blr is required for BBB crossing.** To identify GBS surface component(s) involved in this process, we first tested known virulence and colonisation factors (Fig. 3a), such as the polysaccharide capsule (acapsular mutant Δ*cpsE*), the haemolytic lipid toxin (non-haemolytic strain Δ*cylE*

and hyper-haemolytic strain *cyl+*), or cell-wall anchored proteins (Δ*srtA*). None of these mutants strongly affected BBB crossing (Supplementary Fig. 3a). We thus tested the contribution of surface lipoproteins which are tethered to the cell membrane by an N-terminal lipid moiety. In Gram+ bacteria, lipoprotein biosynthesis involves two specific enzymes, Lgt (prolipoprotein diacylglyceryl transferase) and Lsp (lipoprotein signal peptidase). In our model, removing either Lgt or Lsp decreased bacterial count within the brain at 24 h post-infection compared to wild-type (WT) GBS, and the double mutant (Δ*lgt/lsp*) displayed an additive drop (Fig. 3b), leading to a strong impairment in GBS brain entry. A significant decrease in GBS translocation into the brain was also demonstrated at 6 h post-infection for Δ*lgt/lsp* mutant (Fig. 3c).

Next, we sought to identify specific GBS lipoprotein(s) involved in BBB crossing. The GBS repertoire consists of 39 putative lipoproteins[32], most of them being substrate-binding proteins of ATP-binding cassette (ABC) transporters. We selected Blr (group B leucine-rich), a His-triad/Leucine-Rich Repeat (LRR) protein[33], as an interesting candidate (Fig. 3a'). LRR domains are classically associated with protein-protein interaction and ligand recognition[34]. Similar LRRs are actually found within the internalin A[35] (InlA) of *Listeria monocytogenes*, a surface protein crucial for the bacterial crossing of the gut barrier[36], albeit seemingly not of the BBB[37,38]. To test the role of *blr* (annotated as *gbs0918*), we deleted the gene in GBS NEM316. We first checked that GBS and its isogenic mutants grew similarly in various rich laboratory media as well as in *Drosophila* culture medium (Supplementary Fig. 3b). Moreover, using SEM, we observed no obvious morphological difference between WT GBS and Δ*blr*, that we found attached to the brain surface and in chains, and displaying biofilm-type matrix on their surface (Fig. 3d, matrix colourised in yellow). However, Δ*blr* displayed a significant decrease in bacterial count in the *Drosophila* larval brain at 24 h post-infection compared to the control WT and complemented (Δ*blr + blr*) strains (Fig. 3b). GBS translocation into the brain was also significantly decreased at 6 h post-infection for Δ*blr* mutant (Fig. 3c). Altogether, these results showed that GBS lipoproteins, and in particular Blr, are key contributors to cross the *Drosophila* larval BBB and enter the brain ex vivo.

Surprisingly, we noticed that infection by Δ*blr* mutant resulted in significant damages to SPG membranes, altered septate junction architecture and increased BBB permeability compared to WT GBS (Fig. 3e and Supplementary Figs. 2d and 3c). These differences were decreased but still remained when culture medium pH was maintained (Fig. 3e and Supplementary Figs. 2d and 3c), and acidification was similar regardless of the bacterial

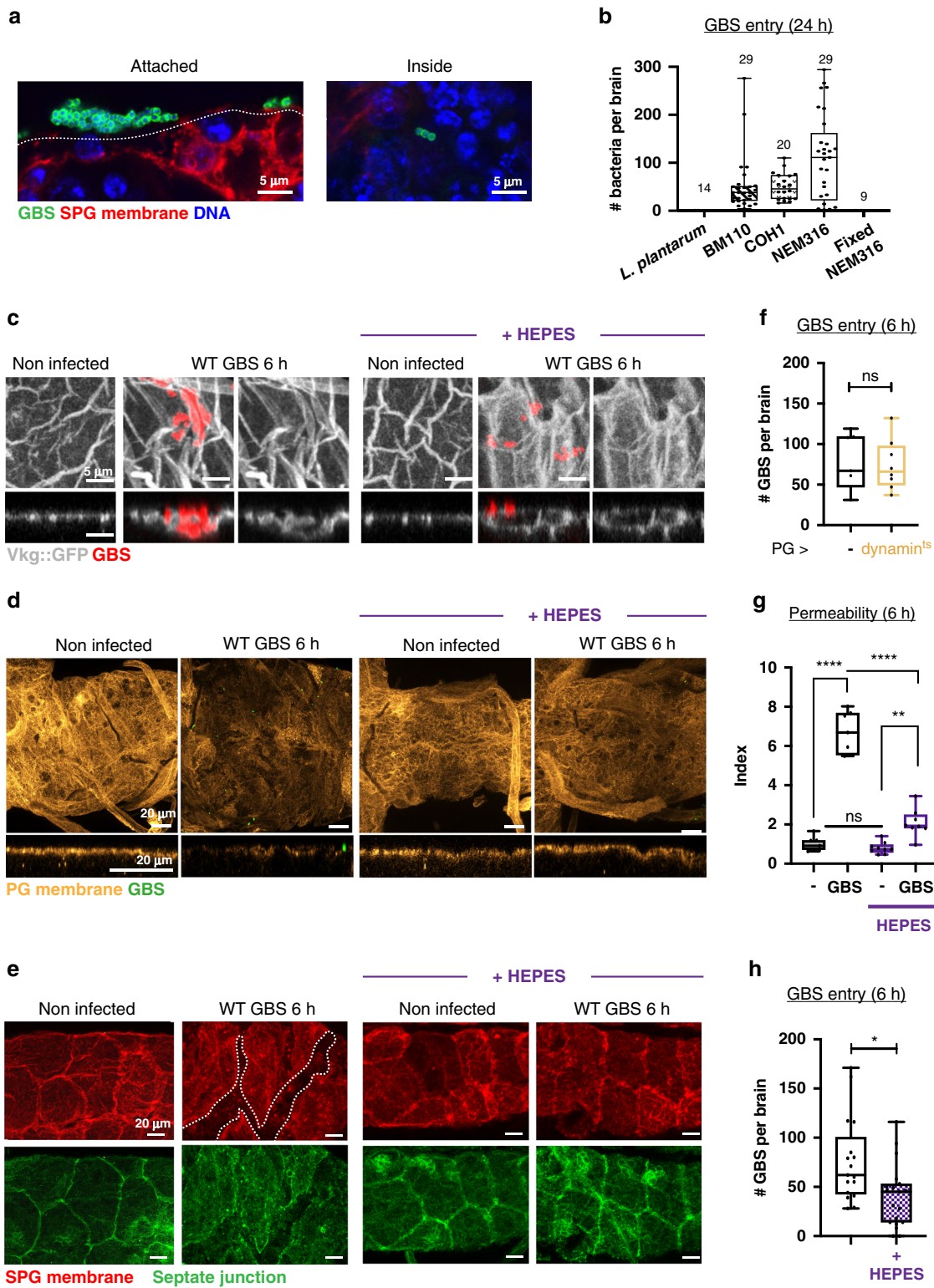

strain (Supplementary Fig. 2f). Of note, none of these features were observed with the double Δlgt/lsp mutant (Supplementary Figs. 2d and 3c). In addition, using SEM, we did not notice a detectable difference in the morphology of Δblr mutants with or without medium acidification (Fig. 3d). Interestingly, we also detected on Δblr-infected brains large film-like structures embedding bacteria and reminiscent of the polysaccharidic coat produced during biofilm formation (Supplementary Fig. 3d, colourised in yellow). Using a marker for polysaccharides (the lectin Concanavalin A, see Methods section), we confirmed that both WT GBS and Δblr mutants were actually able to form biofilms on the Drosophila larval brain (Supplementary Fig. 3e). Altogether, these data suggest that, in the absence of Blr, GBS turns on more destructive, yet much less efficient alternative mechanisms. They also point to specific Blr-dependent mechanisms for GBS crossing of the BBB.

**Fig. 2 GBS uses a panel of strategies to cross the multiple layers of the BBB. a** Close-up of GBS (anti-GBS, green) attached to the SPG (*mdr65-mtd-Tomato*, red) and inside the brain. DAPI, blue. The dashed line outlines the interface between the SPG and the external milieu. **b** Bacterial count inside the brain 24 h post-infection. Numbers above the boxes represent the number of larvae analysed. **c** Close-up of confocal images (top view and orthogonal view) of non-infected and GBS-infected brains at 6 h post-infection, without and with HEPES, showing Collagen IV staining (*Vkg::GFP*, green) and GBS (red). **d** Confocal images (top view and close-up orthogonal view) of non-infected and of GBS-infected brains without and with HEPES at 6 h post-GBS infection, showing PG membrane (*NP6293-GAL4>mCD8-GFP*, light orange). GBS (green). **e** Close-up of confocal images of non-infected brain and brain infected with WT GBS with and without acidosis, showing the SPG membrane (*mdr65-mtd-tomato*, red) and septate junctions (*Lachesin::GFP*, green) at 6 h post-infection. Dashed lines outline SPG damages. **f** GBS entry at 6 h post-infection was not significantly changed when endocytosis was blocked (dynamin$^{ts}$) in the PG. Student's *t*-test: $p = 0.8484$. Control ($n = 7$); PG>dynamin$^{ts}$ ($n = 8$). **g** BBB permeability tests for non-infected ($-$) and GBS-infected brains at 6 h post-infection without (black) and with HEPES (purple). One-way ANOVA test followed by Tukey's multiple comparisons test generated adjusted *p*-values: $p$(- vs WT GBS) $< 10^{-10}$; $p$(- HEPES vs WT GBS HEPES) $= 0.0022$; $p$ (WT GBS vs WT GBS + HEPES) $< 10^{-10}$. $n(-) = 7$, $n$(WT GBS) $= 7$, $n$(- HEPES) $= 8$, $n$(WT GBS + HEPES) $= 8$. **h** Bacterial count inside the brain in GBS entry without (black) and with HEPES (purple) at 6 h (two-tailed Mann–Whitney test, $p = 0.0102$). $n$(GBS) $= 17$ and $n$(GBS + HEPES) $= 19$. Box and whiskers plot: whiskers mark the minimum and maximum, the box includes the 25th–75th percentile, and the line in the box is the median. $*p \leq 0.05$; $**p \leq 0.01$; $****p \leq 0.0001$; ns, not significant. Source data are provided as a Source Data file for **b** and **f–h**.

**The *Drosophila* lipoprotein receptor LpR2 is essential in the BBB for brain invasion by GBS**. We then asked how the lipoprotein Blr overcomes the physical barrier of the SPG (Fig. 1a–c). Interestingly, the LRR-containing InlA was shown to interact with human E-cadherin[39]. We tested the role of the *Drosophila* E-cadherin (*shotgun* gene, *shg*) in GBS entry ex vivo, and found that specifically knocking it down in the SPG layer, through the GAL4/UAS system[40], did not affect GBS brain entry (Fig. 4a).

Lipoprotein receptors were originally identified as surface receptors capable of mediating cellular lipid uptake[41]. Lipids circulate in the blood in association with apolipoproteins, forming particles of different densities (low density, LDL; very low density, VLDL). Lipoprotein receptors are classified into two main groups, based on whether they behave as endocytic receptors supporting lipoprotein internalisation (LDLR, VLDLR, SR-A) or as mediators of lipid exchange at the cell surface. *Drosophila* lipoproteins and lipoprotein receptors are similar to those in vertebrates[42,43]. *Drosophila* has seven lipoprotein receptors, belonging to the (V)LDLR families. Interestingly, specific lipoprotein particles were shown to cross the larval BBB, in which the receptors LRP1 and Megalin are expressed[18,44,45].

We investigated which lipoprotein receptors were important for GBS entry. In addition to previous studies on LRP1 and Megalin in the BBB[45], published transcriptomics data[46] (FlyAtlas available through FlyBase[47], release FB2020_05) suggested expression of *LpR2* and *arr* in the larval nervous system. Specific knockdown of these candidates showed that *LpR2* knockdown in the SPG only was sufficient to decrease GBS count in the brain at 24 h post-infection (Fig. 4a, b). Interestingly, knocking down the closely related and partially redundant *LpR1*[43] did not decrease GBS entry. In accordance with our previous findings (Fig. 2f), LpR2 was not required in the PG layer for GBS brain entry (Supplementary Fig. 4a). To assess LpR2 expression in the larval brain, we used an endogenous LpR2-GFP fusion (*LpR2::GFP* MiMIC line) resulting from a gene knock-in. This line recapitulated previously known profiles of expression in other tissues (wing disc and egg chambers[43], Supplementary Fig. 4b) and behaved in a wild-type fashion for GBS entry (Supplementary Fig. 4c). We observed that LpR2::GFP strongly colocalised with a membrane marker for the SPG (Fig. 4c), a pattern lost upon RNAi-mediated knockdown of *LpR2* in the SPG only (Supplementary Fig. 4d). A similar expression in the SPG was detected using an anti-LpR2 antibody (Supplementary Fig. 4e–f). Interestingly, we were not able to detect LpR2::GFP in the SPG of adult CNS (Fig. 4d), a striking result underlying the existence of different possible mechanisms depending on the life stage. In summary, these results show that LpR2, a lipoprotein receptor specifically expressed in the SPG, is crucial for GBS dissemination into the developing *Drosophila* brain.

**GBS surface lipoprotein Blr binds to the *Drosophila* LpR2, allowing the endocytosis-dependent transcellular crossing of the BBB**. Interestingly, LpR2 has been shown to be an endocytic receptor, able to mediate the uptake of lipoprotein particles[43,48]. We hypothesised that binding of LpR2 to Blr could first help GBS adheres to the SPG, and ultimately lead to its internalisation through endocytosis.

We first tested whether LpR2 and Blr were able to physically interact. We set-up a co-immunoprecipitation experiment between the two species, incubating bacterial lysate on LpR2-GFP fusions extracted from larval brains and bound to beads (see Methods section). We showed that Blr was found in the bacterial eluates from LpR2::GFP beads for wild-type and complemented ($\Delta blr + blr$) strains, whereas no band was recovered from $\Delta blr$ eluates (Fig. 4e and Supplementary Fig. 4g, h). *Drosophila* LpR2 is thus able to bind streptococcal Blr. We next asked whether the LRR domain of Blr was essential for such interaction. We generated a GBS mutant in which the LRR region was deleted ($blr^{\Delta LRR}$, see Methods section and Supplementary Fig. 4i). Using the same co-immunoprecipitation strategy, we found that LRR-deleted Blr was still able to bind to LpR2::GFP, ruling out a strict requirement of LRR region for this interaction (Supplementary Fig. 4j). Altogether these data demonstrated that Blr is able to bind LpR2 in an LRR-independent manner.

We further assessed the role of the endocytic pathway in GBS entry. We blocked endocytosis specifically in the SPG by preventing dynamin function (*shibire*$^{ts}$ and dominant-negative *shibire*$^{DN}$). This led to a strong decrease in bacterial counts within the brain at 6 h post-infection (Fig. 4f). In addition, we were able to detect GBS in vesicles co-staining for a marker of early endosomes (Rab5-GFP) and SPG membrane (Fig. 4g). Expressing another early endocytic marker (FYVE-GFP) specifically in the SPG gave similar results (Supplementary Fig. 4k). In addition, we detected GBS in lysosomal vesicles, coming from the SPG layer, through the specific expression of Lamp1-GFP (Supplementary Fig. 4l) or of Spinster-RFP (Supplementary Fig. 4k, which also marks late endosomes[49]). Finally, we found that GBS and LpR2::GFP colocalised in vesicles staining for SPG membranes (Fig. 4h). Taken together, these results strongly indicate that SPG crossing by GBS occurs via endocytosis, likely through binding of Blr to LpR2 and internalisation of the resulting complexes.

**Blr is a virulence factor essential for BBB crossing in the *Drosophila* larva**. To confirm the relevance of these findings in an

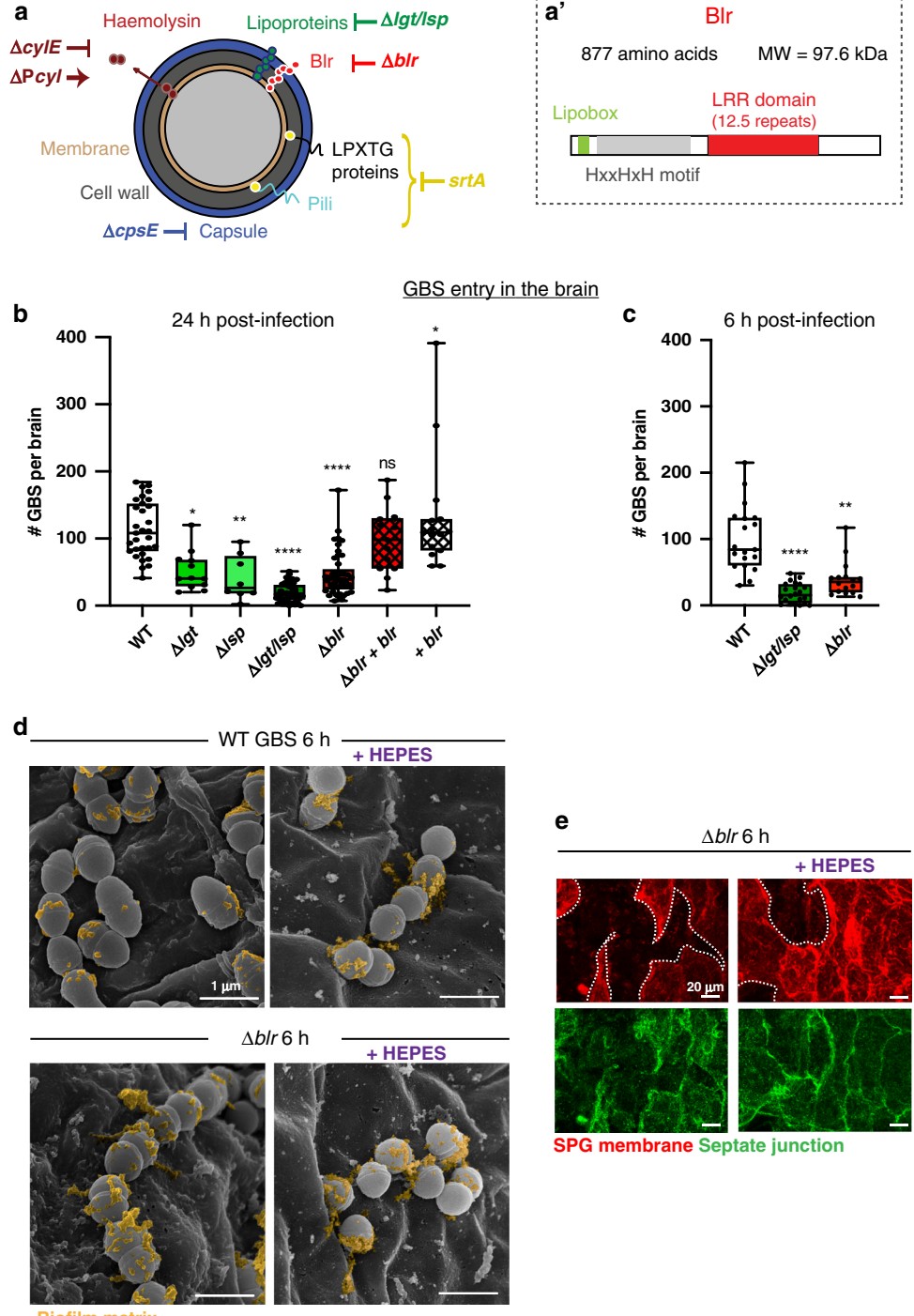

**Fig. 3 Screening for surface factors identifies the lipoprotein Blr as essential for BBB crossing in *Drosophila*. a** Schematic representation of GBS surface structures and tested virulence factors with corresponding mutants. **a'** Schematic structure of Blr lipoprotein. **b, c** Screening of GBS surface structures and virulence factors at **b** 24 h and **c** 6 h post-infection identified GBS surface lipoproteins, and in particular Blr, as crucial for BBB crossing. A Kruskal–Wallis test followed by Dunn's multiple comparisons test generated adjusted *p*-values. **b** WT GBS (*n* = 31) is compared to Δ*lgt* (*p* = 0.0124, n = 12), Δ*lsp* (*p* = 0.0022, *n* = 8), Δ*lgt/lsp* (*p* < 10⁻¹⁰, *n* = 43), Δ*blr* (*p* = 9.77 * 10⁻⁷, *n* = 45), Δ*blr* + *blr* (*p* > 0.9999, *n* = 13), and + *blr* (*p* > 0.9999, *n* = 15). **c** WT GBS (*n* = 19) is compared to Δ*lgt/lsp* (*p* = 1.27 * 10⁻⁸, *n* = 22), Δ*blr* (*p* = 0.0029, *n* = 16). Results are presented as box and whisker plots: whiskers mark the minimum and maximum, the box includes the 25th–75th percentile, and the line in the box is the median. *n* represents the number of larvae analysed. *\*p* ≤ 0.05; \*\**p* ≤ 0.01; \*\*\*\**p* ≤ 0.0001; ns, not significant. **d** SEM pictures of WT GBS and Δ*blr* GBS attached to the brain surface, without or with HEPES. Colourisations show biofilm-type matrix (yellow) present at the surface of the bacteria. **e** Close-up of confocal images of brain infected with Δ*blr*, with and without acidosis, showing the SPG membrane (*mdr65-mtd-tomato*, red) and septate junctions (*Lachesin::GFP*, green) at 6 h post-infection. Septate junctions are strongly affected under Δ*blr* infection without HEPES and partially rescued with HEPES (compare to Fig. 2e). SPG membranes are still damaged under Δ*blr* infection with HEPES (6 h post-infection). Dashed lines outline SPG damages. Source data are provided as a Source Data file for **b, c**.

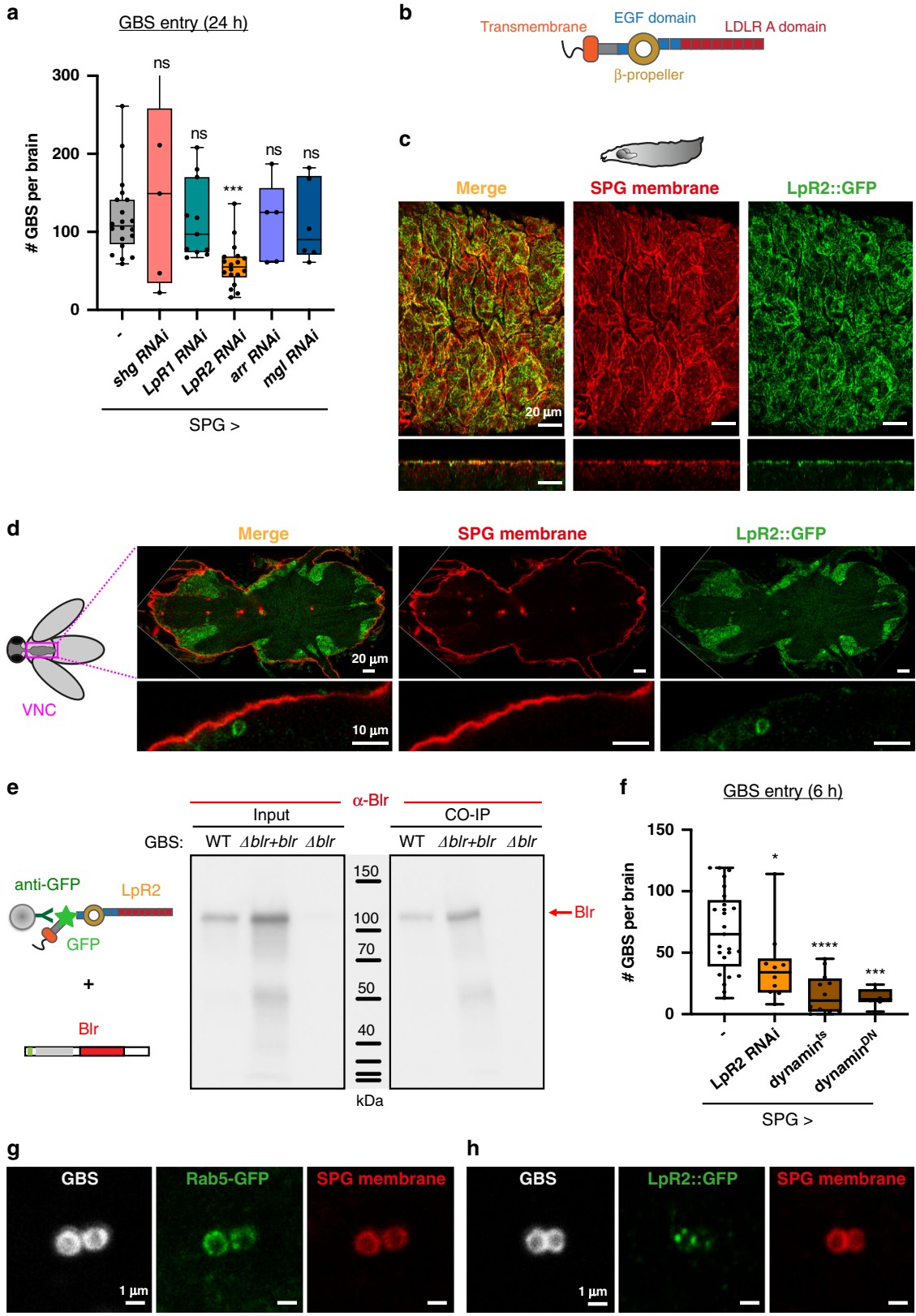

in vivo set-up, we developed a protocol of brain infection through pathogen microinjection into the *Drosophila* circulatory system (Fig. 5a). It was preferred to feeding in order to control the dose and bypass the variability in gut crossing efficiency.

Bacterial counts in the brain of surviving larvae at 4 h post-injection revealed that GBS was able to access and enter the *Drosophila* brain via the systemic route (Fig. 5b). We were also able to observe an altered SPG layer in brains with high bacterial counts (Supplementary Fig. 5a). Survival curves showed that all infected animals died between 4 and 5 h post-injection while mock-injected animals could pass developmental stages and reach adulthood (Supplementary Fig. 5b). These results demonstrated

**Fig. 4 *Drosophila* lipoprotein receptor LpR2 mediates transcellular passage of the SPG by GBS through endocytosis. a** A knockdown screen for *Drosophila* E-cadherin (Shotgun) and lipoprotein receptors (LpR1, *LpR1*; LpR2, *LpR2*; Arrow, *arr* and Megalin, *mgl*) identified LpR2 as crucial for BBB crossing by GBS. A Kruskal–Wallis test followed by Dunn's multiple comparisons test generated adjusted *p*-values. Control (*n* = 20); *shg*, *p* > 0.9999 (*n* = 5); *LpR1*, *p* > 0.9999 (*n* = 11); *LpR2*, *p* = 0.0003 (*n* = 18); *arr*, *p* > 0.9999 (*n* = 5); *mgl*, *p* > 0.9999 (*n* = 6). *n* represents the number of larvae analysed. **b** Schematic representation of LpR2 structure. **c** Confocal image (top and orthogonal views) of LpR2::GFP genomic knock-in line (green) showing colocalisation of LpR2 on SPG membranes (*mdr65-mtd-Tomato*, red) in a larval brain. **d** Confocal image (median cut and orthogonal close-up) showing a lack of colocalisation between LpR2::GFP and SPG membranes (*mdr65-mtd-Tomato*, red) in an adult ventral nerve cord. LpR2::GFP was also detected in neurons. **e** Co-immunoprecipitation experiment between LpR2::GFP immobilised on beads and bacterial lysates of WT GBS, (Δ*blr* + *blr*) GBS and Δ*blr* GBS, detected with an antibody against Blr. A robust Blr-LpR2 interaction was revealed. **f** GBS brain invasion is endocytosis-dependent. GBS entry at 6 h post-infection was significantly decreased by either knocking down *LpR2* or blocking endocytosis (dynamin$^{ts}$ and dynamin$^{DN}$) specifically in the SPG. Two-tailed Mann–Whitney tests were performed between control and each condition: *p*(control vs SPG > LpR2 RNAi) = 0.0162; *p*(control vs SPG > dynamin$^{ts}$) = 4.25 * 10$^{-6}$; *p*(control vs SPG > dynamin$^{DN}$) = 1.04 * 10$^{-5}$. Control (*n* = 25); SPG > LpR2 RNAi (*n* = 10); SPG > dynamin$^{ts}$ (*n* = 12) and SPG > dynamin$^{DN}$ (*n* = 7). **g, h** Colocalisation of GBS (white) with **g** a marker for early endosome (Rab5-GFP) and **h** LpR2::GFP (green) within the SPG membrane (*mdr65-mtd-Tomato*, red). Box and whisker plots: whiskers mark the minimum and maximum, the box includes the 25th–75th percentile, and the line in the box is the median. *n* represents the number of larvae analysed. *\**p* ≤ 0.05; \*\*\**p* ≤ 0.001; \*\*\*\**p* ≤ 0.0001; ns, not significant. Source data are provided as a Source Data file for **a** and **e**, **f**.

that GBS is able to infect the *Drosophila* brain from a circulating, systemic route, causing animal mortality.

Next, we tested the virulence of Δ*lgt/lsp* and Δ*blr* mutants in this set-up. First, bacterial counts in the brains of surviving larvae injected with Δ*lgt/lsp* or Δ*blr* were significantly reduced compared to WT or complemented (Δ*blr* + *blr*) GBS strains at 4 h post-injection (Fig. 5b). To discard differences in fitness or survival between these isogenic GBS strains, we determined through cfu (colony-forming units) counts the exact quantity of bacteria per animal: in or attached to the brain, in the haemolymph, and in all other solid tissues (Supplementary Fig. 5c). We then calculated three ratios: brain to haemolymph, brain to tissues, brain to haemolymph and tissues (Supplementary Fig. 5d). In all cases, we found a significant decrease in Δ*blr* ratios vs wild-type ratios. This shows that the loss of Blr specifically affects the neurotropic ability of GBS to adhere and/or enter the brain. In agreement with these results, survival scores (0–4 h post-injection) were significantly higher in larvae injected with Δ*lgt/lsp* or Δ*blr* mutants compared to the two control strains, with a lethality level similar to non-infected animals (Fig. 5c).

We then assessed the role of LpR2 in the BBB during systemic infection. Infection by WT GBS of larvae in which LpR2 was specifically depleted in the SPG resulted in a dramatic reduction of bacterial count in the brain (Fig. 5d), showing that LpR2 is also crucial for GBS entry into the brain in vivo. Survival curves showed that depleting LpR2 in the SPG did not significantly alter lethality compared to wild-type animals (compare black and orange curves in Fig. 5e). This suggests that, although lethality might result from a brain infection, it mainly depends on a systemic effect and the infection of other organs and compartments.

**Blr is a virulence factor essential for BBB crossing in mice**. To determine whether Blr-dependent virulence and CNS invasion mechanism are conserved in mammals, we used the mouse model of GBS hematogenous brain infection[50] and compared wild-type GBS strain with the isogenic Δ*blr* mutant.

Time-course infection analysis showed that GBS could be detected in the brain as early as 3 h post-infection, was maintained at similar levels at 6 and 24 h, and reduced at 48 h (Fig. 6a). In parallel, bacterial counts in the blood were measurable at 3 and 6 h post-infection and dropped sharply at 24 h (Fig. 6b). Using a fluorescent GFP-tagged GBS, we observed bacteria attached to and in the capillaries of the brain parenchyma at 4 h post-infection (Fig. 6c and Supplementary Fig. 6a) suggesting that the primary entry point for GBS is through the endothelial barrier. Interestingly, we were able to detect LDLR on mouse brain capillaries (stained with CD31),

underlying the availability of this receptor at GBS putative point of entry (Supplementary Fig. 6b). Then, at 24 h after infection, we detected bacteria at the choroid plexuses and walls of the ventricles, including the lateral ventricle (Fig. 6d), that also play a barrier role in the mammalian brain. Very few cells were detected in the brain parenchyma, in regions far from the ventricles, except for some small clusters in which typical streptococcal chains were identified (Fig. 6d).

Survival curves showed that infection with wild-type GBS led to more than 50% of lethality over 7 days (Fig. 6e). The mice that survived up to 7 days exhibited aberrant behaviour indicative of neurological deficits, including unilateral palsy, immobilisation, and imbalance. Mood aberrations, such as isolation and lack of explorative behaviour, were also observed. Moreover, the brains of these mice revealed meningitis hallmarks including meningeal thickening and leukocyte accumulation in the meninges compared with saline-injected control mice (Supplementary Fig. 6c), as identified by co-staining for macrophages (CD68, pan-macrophage marker) and microglia (Iba-1, microglia/macrophage marker).

In contrast, no deaths were recorded in mice infected with Δ*blr* mutant and their survival curve was significantly different compared to mice inoculated with WT GBS (Fig. 6e). We then analysed bacterial levels in the brain and in the blood over the course of infection. The levels of the Δ*blr* mutant in the blood were not significantly different from WT GBS neither at 3 h nor at 6 h post-infection and we observed a similar clearance at 24 h (Supplementary Fig. 6d). Importantly, the brain levels of the Δ*blr* mutant at 3 h and at 6 h were lower, yet not significantly (Supplementary Fig. 6e). A significant reduction was then observed at 24 h post-infection when compared with the WT strain. Normalising brain-to-blood levels confirmed that the Δ*blr* strain was significantly altered in its capacity to invade the mouse brain at 3 and 6 h post-infection, as compared to the WT (Fig. 6f).

Interestingly, none of the mice infected with the Δ*lgt/lsp* mutant died (Supplementary Fig. 6f). Bacterial levels of Δ*lgt/lsp* mutant were reduced both in the blood and brain at 6 h post-infection as compared to WT GBS (Supplementary Fig. 6g). Yet, the brain-to-blood ratios were not significantly different between these two strains (Supplementary Fig. 6h) suggesting that Δ*lgt/lsp* mutants are generally less fit in vivo.

Altogether, these results identify Blr as a new, conserved virulence factor endowing GBS the ability to cross the BBB in *Drosophila* and mouse.

## Discussion

Here we propose an original model of brain infection, using the *Drosophila* larval brain, as a mean to investigate molecular and

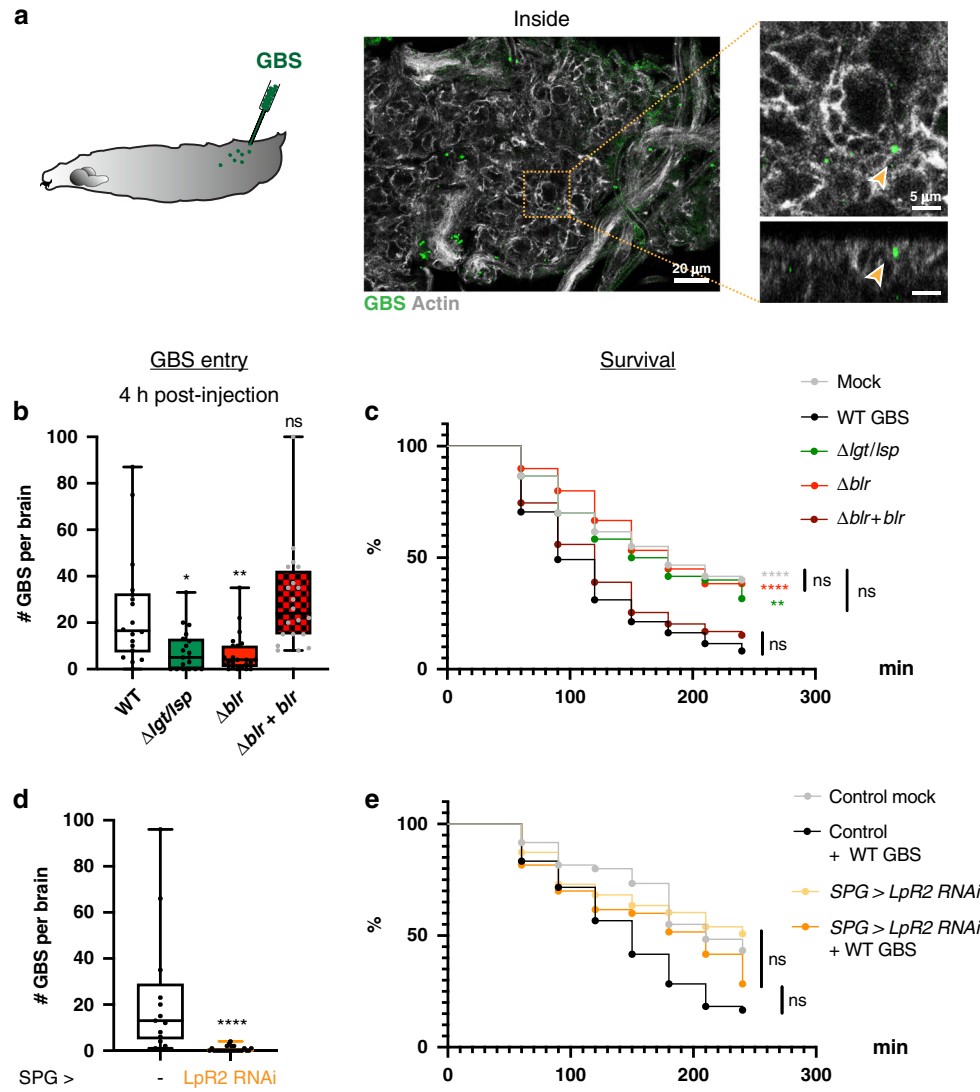

**Fig. 5 An in vivo model of brain infection in *Drosophila* identifies Blr as a virulence factor and confirms LpR2 as a BBB receptor for brain invasion by GBS. a** Schematic representation of *Drosophila* third instar larva injected with GBS. Confocal picture and close-up (top and orthogonal views) showing GBS (in green) inside the brain, 4 h after microinjection. **b** GBS brain entry 4 h post-injection for WT GBS ($n = 18$), $\Delta lgt/lsp$ ($n = 19$), $\Delta blr$ ($n = 23$) and $\Delta blr + blr$ ($n = 20$). A Kruskal–Wallis test followed by Dunn's multiple comparisons test generated adjusted *p*-values: $\Delta lgt/lsp$ $p = 0.0168$, $\Delta blr$ $p = 0.0039$, $\Delta blr + blr$ $p = 0.6579$. **c** Kaplan–Meier survival curves for larvae injected with mock, WT GBS, $\Delta lgt/lsp$, $\Delta blr$ and $\Delta blr + blr$ strains ($n = 60$ for each condition) show that $\Delta lgt/lsp$ and $\Delta blr$ are avirulent. Log-rank test: $p$(WT GBS vs mock) <0.0001; $p$(WT GBS vs $\Delta lgt/lsp$) = 0.001; $p$(WT GBS vs $\Delta blr$) <0.0001; $p$(WT GBS vs $\Delta blr + blr$) = 0.7308, and $p$(mock vs $\Delta blr$) = 0.9686. **d** GBS brain entry at 4 h post-injection in control ($n = 13$) and *LpR2* knockdown ($n = 24$) larvae. Two-tailed Mann–Whitney test: $p = 4 * 10^{-8}$. **e** Kaplan–Meier survival curves for control larvae and larvae in which *LpR2* has been knocked down in the SPG (*SPG > LpR2 RNAi*), injected with mock or WT GBS ($n = 60$ for each condition). Log-rank test: $p$(Control + WT GBS vs *SPG > LpR2 RNAi* + WT GBS) = 0.16; $p$(*SPG > LpR2 RNAi* vs *SPG > LpR2 RNAi* + WT GBS) = 0.13. For **c** and **e**, log-rank *p*-values were adjusted through stacked *p*-values analysis by the Holm–Sidak method. Kaplan–Meier curves show error bars corresponding to standard errors (SE). For results presented as box and whisker plots: whiskers mark the minimum and maximum, the box includes the 25th–75th percentile, and the line in the box is the median. *n* represents the number of larvae analysed. *$p \leq 0.05$; **$p \leq 0.01$; ****$p \leq 0.0001$; ns, not significant. Source data are provided as a Source Data file for **b**–**e**.

cellular mechanisms contributing to the crossing of the BBB. Our model combines an ex vivo approach with brain explants for the straightforward, versatile and scalable screening of putative virulence factors and associated mechanisms, with a full in vivo approach to assessing virulence and impact on the whole organism. Even though the ex vivo protocol does not allow to assess the contribution of circulating immune cells in BBB crossing, bypassing it can unveil BBB-specific mechanisms that could be masked either by an earlier, systemic effect (e.g. general inflammation) or by the difficulty to detect or assess it (e.g. acidosis). Interestingly, for example, *Cryptococcus neoformans* cannot enter the *Drosophila* larval brain in the ex vivo conditions

(Fig. 1e), a finding congruent with the contribution of the Trojan horse mechanism proposed to explain *C. neoformans* barrier crossing[51]. It is worth noting that fly experiments were performed at 30 °C, and not at 37 °C, the usual environment of mammalian pathogens, to allow *Drosophila* development. This constitutes a limitation of our model since the expression of some virulence factors can be temperature-dependent.

Using our model, we aimed to identify novel factors crucial for BBB crossing by GBS. Our approach demonstrated for the first time the contribution of surface-exposed lipoproteins in mediating GBS entry into the *Drosophila* larval brain, and in particular the role of a specific lipoprotein known as Blr. Blr was

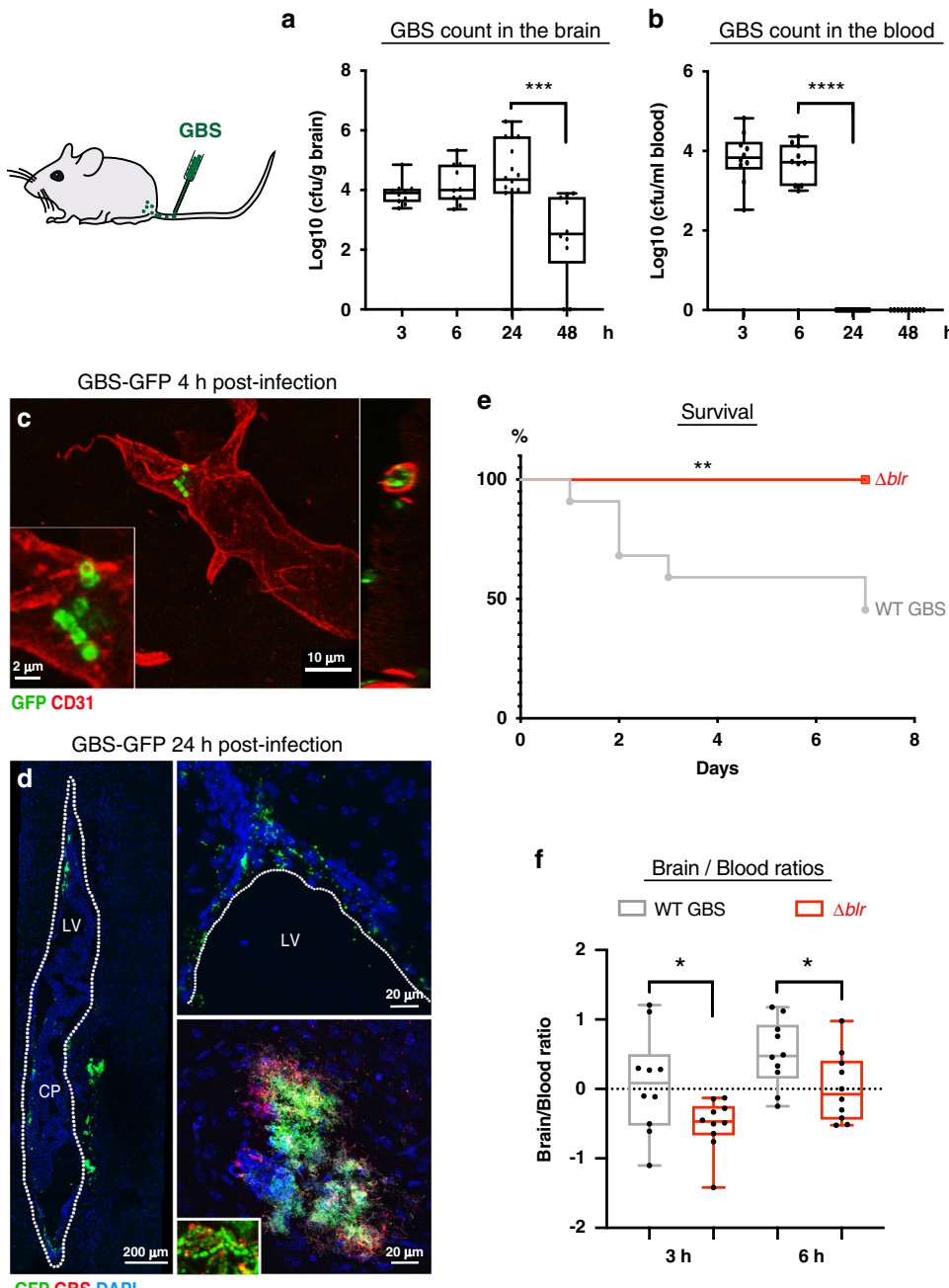

**Fig. 6 Blr is a streptococcal virulence factor in mice involved in BBB crossing by GBS. a**, **b** GBS counts in **a** the brain (including bacteria found in the parenchyma and inside the capillaries) [log10(cfu/g)] and **b** the blood [log10(cfu/ml)] of mice inoculated with WT GBS at 3 h ($n = 10$), 6 h ($n = 10$), 24 h ($n = 18$), and 48 h ($n = 10$). One-way ANOVA followed by Sidak's multiple comparisons tests: $p$(Brain 24 h vs 48 h) $= 0.0003$, and Kruskal–Wallis followed by Dunn's multiple comparisons test: $p$(Blood 6 h vs 24 h) $= 8.5 * 10^{-6}$. **c** Confocal images showing GBS-GFP (green) attached to and in the capillaries (CD31, red) of the brain parenchyma at 4 h post-injection. **d** Confocal images of sagittal brain sections of mice injected with a fluorescent GBS WT-GFP strain showing GFP-positive bacteria (green) at the choroid plexus (CP) inside the lateral ventricle (LV; outlined; left image) as well as at the walls of the LV and in the brain parenchyma adjacent to the LV (upper right image), at 24 h post-infection. In the lower right image, a representative cluster of GFP-positive bacteria (also positive for anti-GBS; red) detected in the brain parenchyma. Typical streptococcal chains found in the clusters are presented in the inset. DNA is stained with DAPI (blue). **e** Kaplan–Meier survival curves of mice intravenously injected with WT GBS ($n = 22$) or $\Delta blr$ ($n = 10$). Log-rank test $p = 0.0055$. **f** The ratio of bacterial counts in the brain vs blood [log10([cfu/g brain]/[cfu/ml blood])] in mice inoculated with $\Delta blr$ was significantly lower than in mice inoculated with WT GBS, at 3 and 6 h post-inoculation ($n = 10$ for each condition). Two-tailed Student's $t$-test, 3 h: $p = 0.0351$; 6 h: $p = 0.0404$. For results presented as box and whisker plots: whiskers mark the minimum and maximum, the box includes the 25th–75th percentile, and the line in the box is the median. *$p \leq 0.05$; **$p \leq 0.01$; ***$p \leq 0.001$; ****$p \leq 0.0001$. Source data are provided as a Source Data file for **a**, **b** and **e**, **f**.

shown to be expressed in vivo but no role in virulence has been demonstrated yet[52]. Interestingly, Blr was shown to bind to the pathogen recognition receptor SR-A (scavenger receptor A), expressed on most macrophages and known to endocytose modified low-density lipoproteins. This finding strongly supports our results that Blr interacts with a specific lipoprotein receptor LpR2 and is then internalised through endocytosis in the SPG. The physiological role of LpR2 in the BBB is not known and the

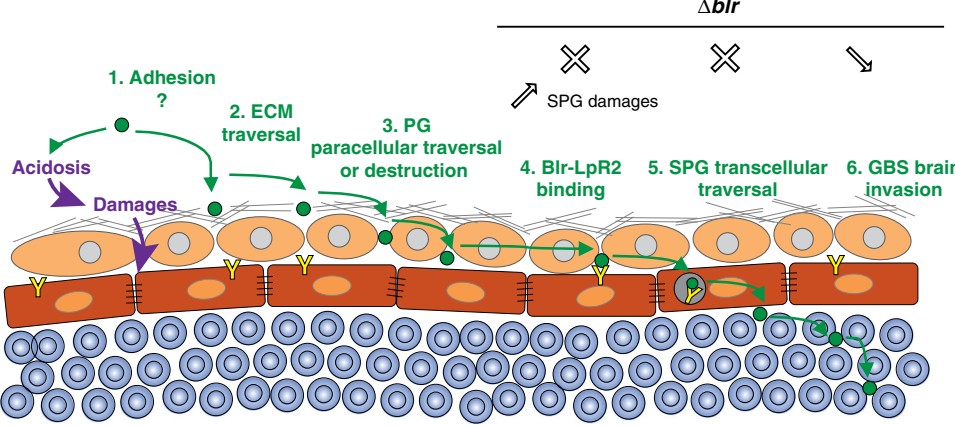

**Fig. 7 Proposed model for the mechanisms used by GBS during BBB crossing.** GBS (green circle) first has to adhere to the ECM layer, before making its way through. It then crosses the PG layer through a paracellular mechanism and/or cellular damages, likely supported or enhanced by acidosis. The bacteria adhere to the SPG surface via Blr-LpR2 (yellow Y) interaction, allowing its internalisation through endocytosis and leading to its transcellular traversal. In the absence of Blr, GBS uses an alternative, albeit less efficient, mechanism for brain invasion, via SPG damages through an unknown process.

exact endocytic journey of the GBS (Blr)-LpR2 complex still remains to be precisely demonstrated.

During GBS infection, some bacterial lipoproteins are released and bind Toll-like receptor 2 through their lipid moiety[53]. However endogenous lipoprotein receptors bind lipoprotein complexes through their protein component (apolipoprotein)[54]. In addition, lipoprotein receptors bind most of their ligands through clusters of cysteine-rich LDL receptor type-A (LA) modules. LpR2, which bears between 7 and 9 LA motifs depending on the isoform[43], could thus bind Blr through its protein moiety. This interaction does not seem to require the LRR domain of Blr (Supplementary Fig. 4j), a surprising result entailing that histidine-triad domain of Blr should be considered as a potential interactor and interesting pharmacological target. Blr is also a virulence factor critical for BBB crossing in mice. *Drosophila* LpR2 is orthologous to mammalian LDLR and VLDLR proteins. Both LDLR and VLDLR were shown to be expressed in brain endothelial cells, where they are linked to the uptake of molecular complexes across the BBB[55,56]. Here we confirmed LDLR localisation in blood vessels of the mouse brain in situ (Supplementary Fig. 6b).

Of note, we found that GBS acidifies the extracellular environment, a known parameter during meningitis. Production of lactic acid by GBS contributes to the weakening of SPG and/or of upstream layers, especially the PG. It has been proposed as a virulence factor in rat fetal lung explants, where it is also linked to tissue destruction[57]. Interestingly, we noticed destroyed blood capillaries in the brain of mice infected with WT GBS (Supplementary Fig. 6a), as well as brains with highly altered SPG during infection by WT GBS in our in vivo *Drosophila* model (Supplementary Fig. 5a). This suggests that acidosis-linked alterations of the BBB might be a conserved mechanism taking place during GBS infection, likely localised around concentrations of bacteria releasing lactic acid. In addition, other events could also account for BBB destruction in vivo.

Surprisingly, Blr-deficient bacteria caused higher damages of the SPG, suggesting that, in the absence of Blr, GBS turn to an alternative pathway, less efficient and more destructive. Such damages were not seen with the lipoprotein-deficient mutant, in which GBS brain entry is extremely low. We hypothesise that other lipoproteins could substitute Blr on the bacterial surface of Δ*blr* mutant, leading to entry into the brain through alternative pathways and thus explaining why Δ*blr* still enters better than Δ*lgt/lsp*. The presence of biofilm is intriguing and could be a way Δ*blr* causes additional damage to the BBB. Altogether, these

different results underline the ability of GBS to shapeshift and use different mechanisms independently or together, depending on the conditions.

How GBS adheres to the *Drosophila* brain is a crucial step that remains to be determined in our model. The ECM is a layer rich in glycosaminoglycans recognised by many pathogens[58], and the fly ECM indeed contains HSPGs, including Perlecan (Supplementary Fig. 2b, c). Moreover, several HSPGs, such as the PG-secreted Dally-like[59], were shown to be important for GBS adhesion to *Drosophila* S2 cells[60] as well as for virulence using an infection model in which adult flies were pricked with GBS serotype Ia (A909 strain)[61].

In conclusion, we propose the following model for GBS entry into the fly developing brain: adhesion, the crossing of the ECM through localised rearrangement, and then traversal of the PG layer, through paracellular and/or destructive mechanisms. Then Blr comes at play, binds to LpR2 on the surface of the SPG allowing GBS endocytosis and brain invasion (Fig. 7). Our work, using an original model of brain infection in *Drosophila*, thus proposes a detailed mechanism behind pathogen crossing of the complex BBB structure and identifies the specific lipoprotein Blr as a new, conserved virulence factor for GBS.

## Methods

**Animal models**. *Drosophila* strains and larval culture conditions

The following fly stocks were used: wolbachia-free $w^{1118}$ (from F. Schweisguth), *mdr65-mtd-tomato* (this study), *mdr65-Gal4* (BDSC 50472[62]), *UAS-mCD8-RFP* (BDSC 27399 and 27400), *NP6293-Gal4* (Kyoto DGGR 105188); *tub-Gal80ts, UAS-shg RNAi* (BDSC stock 34831), *UAS-LpR1 RNAi* (BDSC stock 106364), *UAS-LpR2 RNAi* (VDRC stock 107597), *UAS-arr RNAi* (VDRC stock 4818), *UAS-mgl RNAi* (VDRC stock 105071), *yw; Mi(PT-GFSTF.1)LpR2^{MI04745-GFSTF.1}(BDSC stock 60219), UAS-shi^{ts}* (BDSC stock 44222), *UAS-shi^{K44A}* (BDSC stock 5811), *yw; EGFP-Rab5[63], UAS-GFP-myc-2xFYVE; UAS-spin.myc-mRFP* (BDSC stock 42716), *vkg:: GFP[27], trol::GFP[28]*.

Embryos were collected for 2–3 h on grape juice egg-laying plates. Equivalent numbers (100) of hatching first instar larvae were transferred to standard food plates at 25 °C or 29 °C (for RNAi knockdown) until mid-third instar larval stage. For the *mdr65-Gal4, UAS-RFP x UAS-shi^{ts}*, hatching first instar larvae were transferred to standard food plates at 18 °C until early-third instar larval stage and transferred then to 30 °C.

**Microorganisms used and culture conditions**. The microorganisms that were tested in our experimental set-up are shown in Table 1. All strains were grown overnight at 37 °C in BHI (Brain Heart Infusion) broth for bacteria or in YPD (Yeast extract Peptone Dextrose) medium for fungi. They were stored at −80 °C in BHI broth containing 20% glycerol for bacteria or in YPD broth containing 30% glycerol for yeast. The only exception was *L. plantarum*, which was grown in de Man, Rogosa and Sharpe (MRS) broth and stored at −80 °C in MRS broth containing 20% glycerol.

**Table 1 Microorganisms and PCR primers used in this study.**

| Strains or primers | Relevant characteristics | Reference |
|---|---|---|
| Bacteria | | |
| *Escherichia coli* | | |
| DH5α™-pEGFP-C1 | DH5α™: F– Φ80ΔlacZΔM15 Δ(lacZYA-argF) U169 recA1 endA1 hsdR17 (rK–, mK+) phoA supE44 λ–thi-1 gyrA96 relA1 | 71 |
| *Lactobacillus plantarum* | | |
| Lp^WJL-GFP | | François Leulier |
| Lp^WJL-mCherry | | François Leulier |
| *Streptococcus agalactiae* | | |
| BM110 | | 72 |
| COH1 | | 73 |
| NEM316 | | 74 |
| NEM316ΔcylE | | 75 |
| NEM316ΔsrtA | | 76 |
| NEM316ΔcpsE | | 77 |
| NEM316Δlgt | | 53 |
| NEM316Δlsp | | 53 |
| NEM316Δlgt/lsp | | 53 |
| NEM316Δblr | See Protocols, Construction of NEM316Δblr mutant and complemented strain | This study |
| NEM316Δblr + blr | See Protocols, Construction of NEM316Δblr mutant and complemented strain | This study |
| NEM316 blr^ΔLRR | See Protocols, Construction of NEM316 blrΔLRR mutant | This study |
| NEM316-GFP | See Protocols, Construction of GFP expressing NEM316 | This study |
| *Streptococcus pneumoniae* | | |
| ST4 | | Shaynoor Dramsi |
| *Neisseria meningitidis* | | |
| 2C4.3-GFP | | Muhamed-Kheir Taha |
| *Listeria monocytogenes* | | |
| EGDe | | Marc Lecuit |
| Yeast | | |
| *Saccharomyces cerevisiae* | MAT a his3Δ1 leu 2Δ0 ura3 Δ0 TPI1-GFP-HI3Mx | 78 |
| *Candida albicans* | CEC4061 ura3Δ- Δimm434/ura3Δ-Δimm434 his1-hisG/his1arg4Δ-hisG/arg4 RPS1/RPS1-Clp10-PTDH3-GFP | Christophe D'Enfert |
| *Candida glabrata* | trp1Δ::PTDH3-GFP-AVAL | Christophe D'Enfert |
| *Cryptococcus neoformans* | H99O-E2-Crimson | Guilhem Janbon |

**Mouse ethics statement**. All animal experiments in this study were carried out in the Department of Animal Models for Biomedical Research of the Hellenic Pasteur Institute in strict compliance with the European and National Law for Laboratory Animals Use (Directive 2010/63/EU and Presidential Decree 156/2013), with the FELASA recommendations for euthanasia and Guide for the Care and Use of Laboratory Animals of the National Institutes of Health. All animal work was conducted according to protocols approved by the Institutional Protocols Evaluation Committee of the Hellenic Pasteur Institute (Animal House Establishment Code: EL 25 BIO 013). License No 6317/27-11-2017 for experimentation was issued by the Greek authorities, i.e. the Veterinary Department of the Athens Prefecture. The preparation of this manuscript was done in compliance with ARRIVE (Animal Research: Reporting of In Vivo Experiments) guidelines.

**Protocols**

*Construction of NEM316Δblr mutant and complemented strain.* In-frame deletion mutant of *blr* in NEM316 was constructed by using splicing-by-overlap-extension PCR[64]. The primers used were the following:

- blr-1Eco 5′ blr-1Eco 5′-TTCTgaattcTGTCGGTGCTGTAATGGAGT-3′ /blr-2 5′-TAGCTCCGTAAAAGATTAGAGTCCTCCATAAATGT-3′

  and

- blr-3 5′-AACATTTATGGAGGACTCTAATCTTTTACGGAGCTA-3′ /blr-4Bam 5′-TTCTggatccAACCCCATGATGTAACACT-3′.

The chromosomal gene inactivation was carried out by cloning blr-1/blr-4 fragment into the thermosensitive shuttle plasmid pG1. Electroporation of the recombinant plasmid in *S. agalactiae* NEM316 strain and the allelic exchange was performed as described[65].

To complement the *blr* mutation in *trans*, the *blr* open reading frame was amplified using:

- pTCVblr-1Bam 5′-TCTCggatccTTATGGAGGACTCATGAAAG-3′

  and

- pTCVblr-9BglII 5′- TCTCgtcgacGATTAATGGTGATGATGACC-3′ primer

and cloned into the plasmid pTCV downstream from the constitutive promoter Ptet. The resulting plasmid pTCVΩPtet-*blr* was then transformed into competent NEM316Δ*blr* strain.

**Construction of NEM316 blr^ΔLRR mutant**. In-frame deletion of the leucine-rich repeat (LRR) region of Blr, corresponding to a deletion of 238 aa (from aa 557 to aa 794) in NEM316 was constructed as previously published[64]. The primers used for the splicing-by-overlap-extension PCR were:

- gbs0918-5 5′-TTCTgaattcCACTACCCCAACAGGTAT-3′ /gbs0918-6 5′-TCTTAGCTACTGCTTCAGGCAATCCTTCTAATAGTGGC-3′

  and

- gbs0918-7 5′-GCCACTATTAGAAGGATTGCCTGAAGCAGTAGCTAAGA-3′ /gbs0918-8bis 5′-TTCTggatccTAAACGTCCTTTACTCCCTG-3′.

The gbs0918-5/ gbs0918-bis PCR fragment was finally cloned into the thermosensitive plasmid PG1 and the resulting plasmid was introduced in NEM316 by electroporation. Deletion of the LRR part was obtained by allelic exchange[65]. The deletion was confirmed by PCR and sequencing on the genomic DNA of the mutants.

**Construction of GFP expressing NEM316**. pMV158GFP is a mobilisable plasmid harbouring the *gfp* gene cloned under the control of the P_M promoter[66]. pMV158GFP^Ery plasmid was constructed by replacing the Tc resistance gene of pMV158GFP by the *ermB* gene by using the Gibson method[67]. Briefly, *ermB* and pMV158GFP were amplified with Erm-1 5′-GAGGGTGAAATATGAACAAAA-3′ and Erm-2 5′- CCCTTAACGATTTATTTCCTCC-3′primers, and pMV158-3 5′-TT TTATATTTTTGTTCATATTTCACCCTCCAATAATGAGG-3′ and pMV158-4 5′-TATTTAACGGGGAGGAAATAAATCGTTAAGGGATCAAC-3′, respectively. pMV158GFP and PCR product were ligated and the resulting pMV158GFP^Ery was used to transform *S. agalactiae* NEM316 strain, applying selection for erythromycin (10 μg/ml).

**Bacterial growth curves**. One ml of overnight bacterial preculture in BHI was washed once in PBS and resuspended at $OD_{600}$ of 2 ml$^{-1}$. Then each culture was diluted in a given medium at 1/40 dilution and 180 μl of this suspension dispensed in 96 well plates in triplicate and absorbance measurements were recorded using a Biotek Synergy 2 microplate reader using Gen5 data analysis software (v.3.03).

**DNA cloning and *Drosophila* transgenics**. A portion of the *mdr65* enhancer (GMR54C07, Flybase ID FBsf0000165529), which drives in the SPG, was amplified from genomic DNA extracted from *mdr65-GAL4* adult flies, with a minimal *Drosophila* synthetic core promoter [DSCP[68]] fused in C-terminal. The *mtd-Tomato* DNA codes for a Tomato fluorescent protein tagged at the N-terminal end with Tag:MyrPalm (MGCCFSKT, directing myristoylation and palmitoylation) and at the C-terminal with 3 Tag:HA epitope. It was amplified from genomic DNA extracted from *QUAS-mtd-Tomato* adult flies (BDSC30005, Chris Potter lab). The two amplicons were joined using the Multisite gateway system[69] to generate a *mdr65^DSCP-mtd-Tomato* construct. The construct was integrated in the fly genome at an attP2 docking site through PhiC31 integrase-mediated transgenesis

(BestGene). Several independent transgenic lines were generated and tested, and one was kept (*mdr65-mtd-Tomato*).

**Culture of *Drosophila* brain explants**. Staged larvae were washed successively in PBS and ethanol 70% v/v in water then transferred in cold *Drosophila* Schneider's Medium in a dissection well. Larvae were cut at around a quarter from the posterior spiracle to minimise damages to motor nerves. The posterior part was discarded and the anterior part was turned inside-out to expose the brain. All larval tissues were kept except for the gut, which is removed to avoid contamination with intestinal symbiotic pathogens. Eight larvae were transferred to one well (24-well cell culture plate: Falcon 353504) and cultured in 750 μl of Culture medium I (*Drosophila* Schneider's medium (Gibco 217200-24) supplemented with 2 mM L-Glutamine (Gibco 25030-032) and 0.5 mM Sodium L-ascorbate (Sigma A4034) at 30 °C and 60% humidity under gentle rotary agitation (275 rpm on a Titramax 100 from Heidolph Instruments). 30 °C was chosen as a compromise temperature allowing *Drosophila* development (although with some potential heat response compared to the more standard 25 °C) while culturing mammalian pathogens closer to their usual environment (37 °C). After 3 h, the Culture medium I is replaced by Culture medium II [Culture medium I supplemented with 1% Fetal Bovine Serum (Sigma F4135)], then the medium was replaced after 3 h and every 10 h, by a fresh Culture medium II. In these conditions, brain explants can be kept for up to 48 h, at 30 °C.

***Drosophila* brain explants infection**. An overnight preculture was set from glycerol stocks in BHI (or in MRS for *L. plantarum*) at 37 °C for bacteria or in YPD at 30 °C for yeast. The bacterial preculture was diluted 1/20 in BHI, and was grown for 2 h 30 min at 37 °C ($OD_{600}$ of around 0.8). The yeast preculture was diluted to $OD_{600} = 0.2$ then grown 5 to 6 h at 30 °C until $OD_{600}$ of 1. A 10x infectious dose is then prepared after pelleting through 5 min centrifugation at 3500×g (at 4 °C), washing each original culture twice in PBS, twice in *Drosophila* Schneider's Medium and then resuspended in 750 μl of Schneider's ($10 \times 10^8$ CFU/ml for *Streptococcus agalactiae, Streptococcus pneumoniae, Listeria innocua* and *Listeria monocytogenes*; $10 \times 10^7$ CFU/ml for *Neisseria meningitidis, Candida albicans* and *Candida glabrata* and $10 \times 10^5$ CFU/ml for *Cryptococcus neoformans*). Pathogen concentration was calculated by $OD_{600}$ correlation (*Streptococcus agalactiae, Streptococcus pneumoniae, Listeria innocua* and *Listeria monocytogenes*: $1 OD_{600} = 8.8 \times 10^8$ CFU/ml; *Neisseria meningitidis*: $1 OD_{600} = 10^9$ CFU/ml; *Candida albicans* and *Candida glabrata*: $1 OD_{600} = 3 \times 10^7$ CFU/ml; *Cryptococcus neoformans*: $1 OD_{600} = 6 \times 10^7$ CFU/ml).

The 10× infectious dose of each pathogen is diluted 1/10 in the brain explant culture medium I to reach the infectious dose ($10^8$ CFU/ml). Brain explants were infected for 3 h at 30 °C and 60% humidity under agitation (275 rpm on a Titramax 100 from Heidolph Instruments). Then, the infected medium was replaced by fresh culture medium II after 3 h and every 10 h.

**Dextran permeability**. Brain explants were kept under agitation (275 rpm) for 30 min in 50 mM of 10 kDa Dextran (Texas Red, lysine fixable, D-1863, Invitrogen) diluted in Culture medium II. Brain explants were then immediately fixed 4 × 5 min (to wash-out excess Dextran) in 4% methanol-free formaldehyde.

Permeability index was quantified using ImageJ (version 2020 2.1.0/1,53c) by calculating the average of the mean pixel intensity of three selected equal-sized areas from each brain and subtracting background intensity.

**DHE assay**. To assess oxidative stress, we performed DHE (dihydroxyethidium) assay following standard procedures[70]. Briefly, dissected brains were incubated for 5 min in 30 μM DHE, washed three times in PBS and then fixed for 8 min in 7% formaldehyde in PBS.

**In vivo *Drosophila* larval infection**. GBS preculture and culture are prepared as described for the ex vivo protocol. 20 nl of concentrated GBS were injected in larvae using the nano-injector Nanoject III (Drummond Scientific) in order to reach $8.8 \times 10^8$ CFU/ml of haemolymph. The injected larvae were kept on standard fly food plates placed in a 30 °C incubator with 60% humidity during scoring. Mock injection results in lethality per se, due to a combination of experimental limits:

   i. unsuccessful healing of the punctured cuticle, which should be sealed by a melanisation spot as witnessed in surviving larvae;
   ii. potential damages to tissues neighbouring the injection point, which are favoured by muscular contraction of the larva during injection;
   iii. potential temperature-induced stress (30 °C).

As all animals injected with WT GBS died between 4 and 5 h post-injection while mock-injected larvae could survive up to adulthood, thus passing several developmental stages, we decided to score our conditions until 4 h post-injection to avoid further variable parameters.

***Drosophila* immunohistochemistry**. Brains were processed and stained according to standard procedures. Briefly, brains of inside-out larvae were fixed for 30 min in 4% methanol-free formaldehyde (ThermoScientific, 28908) at room temperature,

washed in PBS 3 × 10 min and permeabilised in PBS-Triton 0.3% for 3 × 10 min. Brains were incubated with primary antibodies at 4 °C in blocking solution (PBS-Triton 0.3%, Bovine Serum Albumin 5%, Normal Goat Serum 2%) for 18–36 h, then washed with PBS-Triton 0.3% and incubated with secondary antibodies 18–24 h at 4 °C in blocking solution, and washed with PBS-Triton 0.3%. The same protocol was used for *Drosophila* adult CNS, using PBS-Triton 1% instead of 0.3%.

Samples were mounted in Mowiol mounting medium and visualised with a laser scanning confocal microscope (Zeiss LSM 880 with Zen software (2012 S4)), with an optimal distance between each slice of 0.38 μm. The following primary antibodies or dyes were used: rabbit anti-GBS (homemade), mouse anti-*S. pneumoniae* (homemade), rabbit anti-*L. innocua* (R6, gift from M. Lecuit), rabbit anti-*L. monocytogenes* (R12, gift from M. Lecuit), chicken anti-GFP (Abcam, ab13970), rabbit anti-LpR2 (gift from J. Culi), Phalloidin–Atto 647N (Sigma 65906), DAPI (Thermo 62247).

Anti-Lpr2 staining is highly variable, and permeabilisation in PBS-Triton 1% was used to help penetration.

Of note, due to medium acidification upon infection, all GFP fusions were detected with an anti-GFP antibody.

**Lectin stainings**. Biotinylated-Concanavalin A (B-1005, Vector Laboratories) was used to stain biofilm polysaccharides. Fixed brains were washed three times 10 min in PBS and incubated 1 h at room temperature with PBS containing 0.1% Tween 20 and 1% BSA. Brains were then incubated overnight at 4 °C with ConA at 1/200 in blocking solution (PBS containing 0.1% Tween 20 and 1% BSA). Brains were then washed with PBS containing 0.1% Tween before 3 h incubation with Streptavidin-A488 at 1/300 in blocking solution (PBS containing 0.1% Tween 20 and 1% BSA). Brains were washed with PBS-Tween 0.1%, mounted in Mowiol mounting medium and visualised with a laser scanning confocal microscope (Zeiss LSM 880 with Zen software (2012 S4)).

**Co-immunoprecipitation and Western blot**. For each condition, 100 brains of *yw; MiMIC(PT-GFSTF.1)LpR2^{MI04745-GFSTF 1}* larvae were dissected and lysed in lysis buffer (50 mM Tris-HCl [pH 7.5], 150 mM NaCl, 1 mM DTT, n-octyl-beta-glucopyranoside 1%, 5 mM EDTA, 1 mM PMSF, protease inhibitor cocktail Roche). Brain lysates were spun for 10 min at 4 °C at 15,000×g and incubated 1 h at 4 °C with 25 μl of equilibrated agarose beads (Chromotek, bab-20) to prevent non-specific binding to beads. The brain lysates were spun for 2 min at 4 °C at 2500×g and the cleared supernatant was incubated overnight at 4 °C with 25 μl of equilibrated GFP-trap beads (Chromotek, gta-20). Bound GFP-trap beads were then washed three times, twice with lysis buffer and once with washing buffer.

Bacterial pellets of different GBS strains *WT, Δblr, complemented Δblr + blr* and *blr^{ΔLRR}* were lysed during 1 h at 4 °C with 1 ml of lysis buffer. The bacterial lysates were spun for 10 min at 4 °C at 15,000×g and the supernatant was incubated 1 h in 25 μl of equilibrated agarose beads at 4 °C. The bacterial lysate was then spun for 2 min at 2500×g at 4 °C and the cleared supernatant was incubated overnight at 4 °C in the column containing bound GFP-trap beads. The column was spun for 2 min at 2500×g at 4 °C and the beads were washed three times, once with lysis buffer and twice with washing buffer. The beads were then resuspended in Laemmli 4× (Bio-Rad) with 10% of β-mercaptoethanol and heated at 90 °C for 10 min.

For Western blot, proteins were boiled in Laemmli sample buffer, separated by SDS-PAGE on 7.5% Mini-Protean TGX Stain-Free precast Gels (Bio-Rad, 4568024), and transferred onto PVDF membrane using the Trans-Blot Turbo transfer pack (Bio-Rad). Immuno-detection was performed as follows: the membrane was blocked in PBS–skimmed milk 5% and incubated for 1 h with rabbit primary anti-Blr[33] (1/750) and rat primary anti-GFP (1/1000, Chromotek [3H9]) antibodies and then with the secondary StarBright_{700}-coupled goat anti-rabbit antibody (1/5000, #12004162 from Bio-Rad) and HRP-coupled goat anti-rat antibody (1/5000, 712-035-153 from Jackson ImmunoResearch). Between the two antibodies and before detection, membranes were extensively washed with PBS + 0.1% Tween 20. Detection. was performed combining fluorescence and chemiluminescence on a Bio-Rad ChemiDoc using Image Lab Software (2020 6.1).

**Scanning electron microscopy (SEM)**. Brains were fixed overnight in 2.5% glutaraldehyde in 0.1 M PHEM buffer pH 7.2. They were washed in 0.1 M PHEM buffer pH 7.2, post-fixed for 1 h and 30 min in 1% osmium tetroxide in 0.1 M PHEM buffer pH 7.2, and then rinsed with distilled water. Samples were dehydrated through a graded series of 25, 50, 75, 95 and 100% ethanol solutions followed by critical point drying with $CO_2$.

Dried specimens were sputtered with 20 nm gold-palladium, with a GATAN Ion Beam Coater and were examined and photographed with a JEOL JSM 6700 F field emission scanning electron microscope operating at 7 Kv. Images were acquired with the upper SE detector (SEI) and using JEOL software module (PC-SEM Main Executable version 3.31.13).

**Transmission electron microscopy (TEM)**. For transmission electron microscopy, brains were fixed with 2.5% glutaraldehyde in 0.1 M PHEM buffer pH 7.2 overnight at 4 °C. Specimens were post-fixed with tannic acid 1% in 0.1 M PHEM buffer pH 7.2 for 30′, post-fixed with 1% osmium tetroxide for 1 h and 30 min in 0.1 M PHEM buffer pH 7.2 at room temperature, dehydrated in a graded series of ethanol, and embedded in Epon. After heat polymerisation, thin sections were cut

with a Leica Ultramicrotome Ultracut UC7' sections (60 nm), stained with uranyl acetate and lead citrate. Images were taken with a Tecnai SPIRIT (FEI-Thermofisher Company at 120 kV accelerating voltage with a camera EAGLE 4 K × 4 K FEI-ThermoFisher Company) using TIA software V4.

**Mouse infection**. Eight to 10-week-old male CD-1 mice (body weight, $40.99 \pm 3.62$ g [mean ± standard deviation]) were randomly grouped and injected intravenously (i.v.), via the tail vein, with $10^8$ CFU of bacterial suspensions in sterile normal saline. A priori sample size estimation was performed using GPower version 3.1. For the determination of bacterial levels in blood and brain, mice were anaesthetised by intraperitoneal (i.p.) injection of a mixture containing ketamine (Imalgene 1000, MERIAL, Lyon, France; 100 mg/kg of body weight) and xylazine (Rompun, Bayer, Leverkusen, Germany; 10 mg/kg of body weight). Blood samples were collected by cardiac puncture. Immediately after, each mouse was killed by cervical dislocation and its brain was aseptically removed. One brain hemisphere from each mouse was homogenised in sterile normal saline. Bacterial levels in blood samples and brain homogenates were determined by plating serial tenfold dilutions on Columbia Agar with Sheep Blood plates (ThermoFisher Scientific, Waltham, MA, USA) and counting of bacterial colonies 16 h later. The numbers of mice in each group of analysis are shown in Table 2. The bacterial loads per animal were then represented in a Log10 scale, and the brain/blood ratios were calculated as follows: ratio brain/ blood = log10 [(cfu/g brain)/(cfu/ml blood)]

**Mouse immunohistology**. Mice were euthanized by (i.p.) injection of a ketamine/xylazine mix. After transcardial perfusion with 4% paraformaldehyde in phosphate-buffered saline (PBS), the brains of the infected mice were dissected out and post-fixed in the same fixative, cryoprotected in 30% w/v sucrose solution in PBS for 2 d at 4 °C, embedded in O.C.T. compound (VWR Chemicals) and frozen at −80 °C.

### Table 2 Sample size per time point per bacterial strain.

| | Blood and brain levels | | | | Survival |
|---|---|---|---|---|---|
| | 3 h | 6 h | 24 h | 48 h | 7 d |
| WT GBS | 10 | 10 | 18 | 10 | 22 |
| Δlgt/Δlsp | | 12 | 17 | 10 | 10 |
| Δblr | 10 | 10 | 9 | | 10 |

Series of coronal or sagittal 20-µm-thick sections were collected on Superfrost Plus microscope slides and stored at −20 °C until further processing. The cryosections were thawed and subjected to antigen retrieval in 10 mM sodium citrate solution, pH 6, followed by 1 h blocking of non-specific sites with 5% v/v normal donkey serum (NDS), simultaneously with permeabilization using 0.1% v/v Triton X-100 in PBS. Primary antibodies diluted in 2.5% NDS in PBS were applied overnight at 4 °C, followed by incubation with the appropriate secondary antibodies for 2 h at room temperature. The following primary antibodies were used: rat anti-Cluster of Differentiation 68 (CD68; 1:100; Bio-Rad Antibodies, Oxford, UK; MCA1957GA), rabbit polyclonal anti-ionised calcium-binding adapter molecule 1 (Iba-1; 1:400; FUJIFILM Wako Pure Chemical Corporation, Osaka, Japan; 019-19741), rabbit anti-CD31(1:50; Abcam, Cambridge, UK; ab28364), goat anti-LDLR (1:100, R&D Systems, MN, USA; AF2255), rabbit anti-GBS (1:300; homemade). Secondary antibodies (all from ThermoFisher Scientific) used for immunofluorescence were conjugated with Alexa Fluor 488 or 546 and cell nuclei were counterstained with 4′,6-diamidino-2-phenylindole (DAPI; 1:1000; ThermoFisher Scientific). Prolong Gold antifade curing mountant (Cell Signaling Technology, Danvers, MA, USA) was used for mounting. Images were acquired using Leica TCS SP8 confocal microscope with Leica Application Suite X software version 3.5.5.

**Image processing**. Fiji (ImageJ version 2020 2.1.0/1,53c and version 1.52p), Icy (2.0.3.0) or Volocity (6.3) were used to process confocal data. Adobe Photoshop and Illustrator were used to assemble Fig.s.

**Statistics and reproducibility**. GraphPad Prism software (version 7 and version 2020 8.4.2 (464)) was used for all analyses.

**Bacterial quantifications in infected *Drosophila* brain**. The same region of the CNS (Ventral Nerve Cord, VNC) was scanned at an optimised number of slices (distance between each slice of 0.38 µm) using a Zeiss LSM 880 microscope with Zen software (2012 S4). The exact number of bacteria for each brain was then determined manually by counting each individual bacterium contained within the boundary of the BBB (*mdr65-mtd-Tomato*).

**CFU counts following in vivo larval GBS-injection**. Each injected larva is washed on a paper with ethanol 70% then bled in 10 µl PBS. The brain is then dissected, transferred and homogenised in 10 µl PBS. The rest of the larval carcass (other tissues except the gut) is also transferred and homogenised in 10 µl PBS. This protocol was done for 5 larvae by the condition. Haemolymph, brain and carcass bacterial levels were determined by plating 7 serial tenfold dilutions two times on

### Table 3 Experimental reproducibility.

| Figure | Total number of samples | Number of experiments | Overall penetrance (%) |
|---|---|---|---|
| 1b | 16 CNS | 2 | 100 |
| 1d | 20 CNS | 2 | 100 |
| 2a | 80 CNS | 12 | 100 |
| 2c | ≥14 CNS per condition | 2 | 100 |
| 2d | ≥12 CNS per condition | 2 | 100 |
| 2e | ≥12 CNS per condition | 3 | 100 |
| 3d | ≥4 CNS per condition | ≥1 | 100 |
| 3e | ≥12 CNS per condition | 3 | 100 |
| 4c | 18 CNS | 4 | 100 |
| 4d | 10 adult CNS | 2 | 100 |
| 4e | NA | 3 | 100 |
| 4g | 8 CNS | 1 | 100 |
| 4h | 13 CNS | 1 | 100 |
| 5a | 13 larvae | 3 | 100 |
| 6c, d | ≥3 mice per condition | 1 (based on the 3R principle) | 100 |
| Supp. 1d | ≥5 CNS per condition | 1 | 100 |
| Supp. 2a | ≥14 CNS per condition | 2 | 100 |
| Supp. 2b, c | 8 CNS per condition | 1 | 100 |
| Supp. 2d | ≥12 CNS per condition | 3 | 100 |
| Supp. 2e | ≥5 CNS per condition | 1 | 100 |
| Supp. 3d | 5 CNS | 2 | 40 (2 CNS) |
| Supp. 3e | ≥7 CNS per condition | 2 | 15–20% (1–2 CNS) |
| Supp. 4b | ≥14 larval VNCs; ≥3 wing discs; ≥3 egg chambers | 4; 2; 2 | 100 for each tissue |
| Supp. 4d | ≥7 CNS per condition | 2 | 100 |
| Supp. 4e | 13 CNS | 2 | 45 (6 CNS) |
| Supp. 4f | 17 CNS | 3 | 40 (7 CNS) |
| Supp. 4g, h | NA | 2 | 100 |
| Supp. 4j | NA | 1 | 100 |
| Supp. 4k, l | 8 CNS per condition | 1 | 100 |
| Supp. 5a | 8 control CNS;13 infected CNS | ≥2 | Control: 100 Infected: 15 (2 CNS) |
| Supp. 6a–c | ≥3 mice per condition | 1 (based on the 3R principle) | 100 |

Columbia Agar with Sheep Blood plates (Biomérieux 43041) and counting of bacterial colonies after 16 h at 37 °C. The average CFU/µl was calculated as an average from all the different dilutions. The bacterial loads per animal were then represented in a log10 scale, and ratios were calculated from raw counting then represented on a log10 scale:

- Ratio brain/haemolymph = log10 (cfu per brain/cfu per haemolymph)
- Ratio brain/other tissues = log10 (cfu per brain/cfu per Other tissues)
- Ratio brain/(haemolymph + other tissues) = log10 (cfu per brain/(cfu per haemolymph + cfu per other tissues)).

**Drosophila statistical analysis**. All *p*-values are exact.

In order to perform statistical tests on several experimental replicates, each value (corresponding to one brain) was normalised to the mean of the control condition within one replicate. Statistical tests were then run on all the normalised values from all replicates, which were considered as biological replicates.

Comparisons between BBB permeability, GBS entry into the brain, cell viability, oxidative stress, bacterial levels in the haemolymph, bacterial levels in other tissues, bacterial levels in the brain, the ratio of bacterial levels for brain/haemolymph, the ratio of bacterial levels for brain/other tissues and ratio of bacterial levels for brain/other tissues + haemolymph were performed by Student's *t*-test (two conditions) or one-way ANOVA test followed by Tukey's post-hoc analysis (more than two conditions) when values followed a normal distribution (assessed by Shapiro–Wilk normality test). Otherwise, non-parametric Mann–Whitney tests (two conditions) or Kruskal–Wallis tests (more than two conditions) were performed. The data were represented with Box and whiskers plots. All Box and whiskers plots display minimal value (bottom whisker), first quartile (25th percentile, lower limit of the box), a median of the interquartile range (middle horizontal line), third quartile (75th percentile, the upper limit of the box) and maximal value (top whisker). All individual points are plotted.

Comparison of survival curves was performed using the log-rank test. The log-rank test is based on a chi-square distribution and tests for the difference between two or more survival curves without any prior on the direction of the difference. When more than two conditions were considered, *p*-values were adjusted by determining their statistical significance (alpha = 0.05) through stacked *p*-values analysis through the Holm–Sidak method. Data were represented as Kaplan–Meier curves with error bars corresponding to standard errors (SE).

*p*-values lower than 0.05 were considered significant.

**Mouse statistical analysis**. Comparisons between bacterial levels in the blood and the brain, as well as between ratios of bacterial levels for brain/blood were performed by unpaired Student's *t*-test or one-way ANOVA followed by Sidak's multiple comparisons test when values followed a normal distribution (assessed by D'Agostino-Pearson normality test). Otherwise, non-parametric Kruskal–Wallis followed by Dunn's multiple comparisons test was performed. The data were represented with Box and whiskers plots. All Box and whiskers plots display minimal value (bottom whisker), first quartile (25th percentile, lower limit of the box), a median of the interquartile range (middle horizontal line), third quartile (75th percentile, the upper limit of the box) and maximal value (top whisker). All individual points are plotted.

Comparison of survival curves was performed using the log-rank test.

*p*-values lower than 0.05 were considered significant.

**Representative pictures**. For representative pictures of phenotypes and experiments, the total number of biological samples and independent experiments, as well as the percentage of samples showing the represented phenotype are displayed in Table 3.

**Reporting summary**. Further information on research design is available in the Nature Research Reporting Summary linked to this article.

## Data availability

The datasets generated during and/or analysed during the current study are available from the corresponding author on reasonable request. Source data are provided with this paper.

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

## Acknowledgements

We thank J. Culi, C. d'Enfert, G. Janbon, F. Leulier, M.-K. Taha, M. Lecuit and F. Schweisguth for reagents and strains. We are very grateful to Gunnar Lindahl for the kind gift of his homemade antibody against Blr lipoprotein. L. Arbogast generated the *mdr65-mtd-Tomato* construct. Stocks obtained from the Bloomington *Drosophila* Stock Center (NIH P40OD018537) and from the Vienna *Drosophila* Resource Center were used in this study. We are grateful to A.-E. Deghmane, O. Disson, C. d'Enfert, G. Janbon, M.-K. Taha and M. Lecuit for their kind and enthusiastic help and advice with the pathogen screen. We are indebted to D. Ferrandon for his expert advice on the project and to S. Liégeois for technical help, especially introducing us to haemolymph injection of *Drosophila* larvae. We thank H. Varet from the Department of Computational Biology of the Institut Pasteur Biostatistical Hub for an enlightening discussion about biosta-tistical analysis, A. Mallet from the UTechS Ultrastructural Bioimaging for advice on electron microscopy and B. Montagne for help with the Western blots. J.-M. Panaud colourised the SEM pictures from Fig. 3d and Supplementary Fig. 3d. We thank E. Voulgari for valuable advice and S. Trygoni and D. Dionysopoulou for technical support on the mouse infection model. Further, we are grateful to the personnel of the Department of Animal Models for Biomedical Research of the Hellenic Pasteur Institute for their invaluable help. We thank D. Ferrandon for critical reading of the manuscript. This work has been funded by a starting package from Institut Pasteur/ LabEx Revive and a JCJC grant from Agence Nationale de la Recherche (NeuraSteNic, ANR- 17-CE13-0010-01) to P.S.; a Grand Projet Fédérateur Microbes & Brains InFeSteR grant from Institut Pasteur to P.S., R.M. and S.D. B.B. has been supported by a Roux-Cantarini (Institut Pasteur) and a LabEx Revive post-doctoral fellowships.

## Author contributions

B.B. performed most *Drosophila* experiments. N.R. generated Figs. 3c, 4c, e–f as well as parts of Fig. 2c and of Supplementary Fig. 2a, d, all under the supervision of B.B. P.S. devised the brain explant protocol and started the pathogen screen. B.B. and P.S. designed and analysed all *Drosophila* experiments. B.P. and S.D. generated GBS strains and constructs as well as performed growth curves. S.D. designed and advised on all GBS experiments. F.P. designed, performed and analysed, together with K.S., the mouse experiments and generated Fig. 6 and Supplementary Fig. 6. V.M. provided consultation on the mouse infection model and resources. R.M. supervised and advised on all mouse experiments and analysis. C.S. generated the EM and SEM data. B.B., S.D. and P.S. wrote the article with input from F.P. and R.M.

## Competing interests

The authors declare no competing interest.
