## [Peer Review File · Nature Communications]

Reviewers' Comments:

Reviewer #1:

Remarks to the Author:

This manuscript is a fascinating new model to examine how bacteria penetrate barriers using fly larvae. Fly genetics allows for more rapid screening and the authors demonstrate in detail the methods to visualize and interpret analyses in such a small organ. The cell biology is strong and explained well. It is highly likely that many labs will try the model and hopefully the field will be advanced by this new experimentation.

Comments:

- 1) Could the authors add in an estimate of how the fly larval stage compares with the mouse developmental day (embryonic day ?)? It is telling that only GBS, a neonatal pathogen, is strong in the model and this may relate to the developmental age of the system; please comment.
 - 2) The discussion could comment on the lack of any previously found determinant to be active in their model. It is highly unlikely that all other studies are wrong. What are the limitations of the model?
 - 3) Figure 5 panels C and E: the inset legends do not match the lines in the graphs. This is currently uninterpretable.
- E. Tuomanen

Reviewer #2:

Remarks to the Author:

Benmimoun et al. use *Drosophila* as a model organism to screen for factors implicated in host-pathogen interactions. They succeed in identifying the route that streptococcus agalactiae uses to pass the blood brain barrier in *Drosophila*. They combine ex vivo as well as in vivo infection setups to analyze molecular interactions between bacterial and fly proteins. Further, they show that the route of entry used in the fly is likely to be conserved in mammals. Overall, I find this study very interesting and recommend its publication. However, I have some concerns that I think need to be considered before publishing the study.

Major concerns:

1. I have some doubts about the disruption of the ECM and the PG cell layer. In Figure 2c The authors show GBS infected brains stained for GBS and vkg::GFP. They claim that these stainings show a disruption of the ECM by GBS. The top view indicates holes in the ECM where the GBS are (is this a single plain?). However, the orthogonal view rather indicates slight bending of the ECM. In this view I cannot see a strong difference between the non-infected and the infected ECM. Thus, I have strong doubts about the destruction of the ECM here.
Further, concerning the damage of the PG layer: Figure 2e shows the expression of a membrane-bound GFP specifically in the PGs of infected, non-infected and infected+HEPES animals. The GFP signals appears much weaker in the infected brains vs. the non-infected brains or the infected, HEPES-treated brains. The authors claim that this change in GFP signal is due to a destruction of the PGs by acidification of the milieu caused by the GBS. Depending on how these pictures were taken, however, I have strong doubts about this interpretation. The GFP usually used in transgenic flies is highly pH-sensitive. It hardly fluoresces at a pH of 5 (Roberts et al. Scientific Reports 2018). Since the pH that was measured in the medium during infection is 4.6 (!) a change in the GFP signal might not be due to destruction of the PGs and thus loss of membrane-bound GFP, but simply to loss of GFP fluorescence, at least if those pictures have been taken live. Since I do not know how fast GFP-fluorescence recovers after exposure to low pH, I suggest that the same experiments should be redone with fixed and anti-GFP stained samples to make sure, that there is really a loss of GFP protein.
2. Effect of acidification on the BBB. The authors claim that the addition of HEPES reverses the destructive effect on the BBB. The data using a permeability assay seem valid, but I have problems with the EM pictures. In figure 2g the authors show an EM picture of wt infected with GBS + HEPES

and claim that the phenotype is reversed. Since there is no picture of wt + GBS without HEPES, I have no idea what the phenotype on EM-level would be. I would strongly suggest adding an EM-picture of wt+GBS. In Figure S2d there is a picture of a non-infected wt, but it has a different magnification. Further there is a zoom of the same picture as in 2g in S2e. I do not know what additional information this should give. I suggest showing EM-pictures of wt non-infected, wt infected and wt infected + HEPES all at the same magnification in Figure 2.

3. I further have a general doubt about the importance of this acidification for in vivo infections. The medium (Schneider's medium) used in the ex vivo infection setup has not much to do with the hemolymph that is filling the body cavity in vivo. Probably also the bacterial load is not similar to in vivo infections. Can the hemolymph in vivo really be that strongly acidified without killing the animal immediately? Is this a realistic scenario or is the idea rather a local acidification there? Maybe it would be a good idea to measure hemolymph pH of infected vs. non-infected animals (in vivo model). In addition: Are there signs of PG/SPG destruction in the in vivo infection model? Further, one could infect ex vivo brains using bacterial mutants that are not able to acidify the medium anymore (unfortunately I am not a specialist on GBS and do not know by which mechanism this happens, and thus cannot suggest something specific).

4. Putative biofilm formation by GBS delta blr. The authors claim that figure S3d shows putative biofilm formation in delta blr infected brains. This is not obvious. What is meant by "additional large structures"? Could you please mark them somehow? In addition, is this also visible in 3e? I would expect that it should be. If this is the case, why are there additional pictures in S3? In addition, as for the EM experiments before, the controls are missing. How does the infected wt look without HEPES? How does delta blr infected look without HEPES? How does figure Sf2 relate to this – I think this figure is not even mentioned in the text. I would recommend presenting the EM pictures in one figure together with the proper controls.

5. LPR2 localization. Figure 4b and figure S4b-c do not agree on the localization of Lpr2. Figure 4b shows staining of an LPR2-GFP protein trap, while S4b-c shows stainings using an antibody against Lpr2 in wt animals(?). The LPR2-GFP protein trap seems to be much more abundant and more evenly distributed along the membrane than suggested by the anti-LPR2 staining. Further, the localization of the anti-LPR2 stained LPR2 at the SPG membrane is not obvious in figure S4b-c. I do not see colocalization with the SPG-membrane marker. I rather get the impression that the staining is found in the PGs. Here, I think better pictures would be important.

Since fusing a GFP to a protein might interfere with its localization, function or stability, it is important to clarify which staining reflects the "wild-typic" situation. As a first step I would suggest costaining Lpr2-GFP animals for GFP and LPR2 to see if the stainings overlap. If this is the case one must assume that the GFP-fusion is more stable/localizes differently than the wt protein and cannot be used to draw any conclusions.

6. In vivo infection model: I have some doubts about the in vivo infection model. Why do 60% of the mock-injected wt larvae die within 4 hours? Such a high mortality rate that is not associated with an infection, seem to be a bad basis for evaluating effects of an infection. Further, I do not understand, why the survivals have not been scored until all (at least the infected) animals are dead? Do any of the mock-injected wt animals survive? At what temperature are these experiments performed? If it is 30°C as for the ex vivo infection, this might be part of why they die, since animals usually are rather stressed at 30°C. Have infections using adult animals been tried? I understand that all other *Drosophila* data comes from larvae and thus a larval infection model makes sense, but if it is so difficult to infect them without killing them independent of the infection, it might make sense to switch to the adult here. Adults survive injections very well. Also, the text states that the mutant bacteria are less virulent than wt bacteria. Having a look at the survival curves, I would say that they are avirulent, since there is no difference between non-infected animals and those infected with mutant bacteria. This also fits the mouse data.

Concerning the Lpr2-RNAi survivals: How has the statistical analysis been done? Were the entire curves compared to each other or just the single time points? I can hardly believe that there should not be a significant difference between wt+GBS and SPG>Lpr2-RNAi GBS (graph says n.s., even though the text states there is a difference - please clarify). On the other hand, it is difficult to believe that there is really a difference between non-infected and infected SPG>Lpr2-RNAi animals. Has a

multiple comparison using Holm-Sidak been done? I would strongly suggest talking to an expert on statistical analysis of lifespan data. Also: the legends do not fit the graphs (figure 5c,e)!

Bacterial counts in the brain are shown for 4h p.i. and 18h p.i. (or 16h as stated in the figure? – please clarify). How many animals are even alive at 18h post infection? Are these then just those animals that have not been properly injected? I have severe doubts that this number tells anything about the infection process.

If I understand the data correctly, Figure 5 b, d and S5a show bacterial counts within the brain based on brain stainings and manual counting of bacteria that are inside the brain. In contrast figure S5b,c are based on bacterial counts as determined in homogenized tissue by plating the homogenate and counting colonies. Thus “brain” in S5b,c includes all bacteria within the brain and attached to the brain. This is an important difference that should be made clear.

The number of bacteria in the brain after 4h of infection is very low. I have difficulties to believe that such a low number of bacteria would kill the animal. This should be discussed.

7. Mouse infection model: The authors claim that there are “destroyed blood capillaries” found in GBS infected mouse brains. They refer to Figure s6a. In S6a one might believe this, however in S6b, which I think shows the same mouse-bacterium combination, this does not seem the case. Are there really destroyed capillaries, or is this due to sectioning? Maybe a 3D reconstruction of the capillaries could help here, compared to non-infected animals of course. If the authors want to show a difference between the capillaries, control stainings of non-infected animals should anyways be shown.

Minor concerns:

1. Acknowledgments: mdr65-mtdt-tomato \diamond mdr65-mtd-tomato
2. Figure 1a: I do not understand why the hemolymph is depicted as spheres instead of filling up the whole animal? Also, the authors talk about “brain cell populations”, but just show neurons and neglect cortex glia, astrocyte-like glia and ensheathing glia.
3. Figure 1d. scale bars are missing. In the orthogonal sections I guess the more actin-rich region is the neuropil. Where is it in the left-most picture. Are the stacks comparable?
4. Figure 2a: Which region is shown? What is the dashed line?
5. Figure 2c: the figure legend says vkg::GFP, the figure says COIIV::GFP. I would suggest sticking to the fly nomenclature here.
6. Figure 2e: the non-infected +HEPES control is missing
7. Figure 3: A and A' should be separated in the legend
8. Figure 4a: Please add gene abbreviations in the legend (shg, arr, mgl)
9. Figure 4c: why is there no anti-BLr blot of the brain lysates (since they are also an “input”) to show that they are negative? Similar in Figure S4d: Why are the bacterial inputs not probed? They should be negative for Lpr2-GFP.
10. Figure 4 e-f: what are the schematic representations on the left? They are not mentioned anywhere (even not in the legend). Please either explain them or take them out. Further, the merge is not very helpful the way it is presented since the “white” channel is too prominent. What is the blue channel in the merges? It is not mentioned anywhere. The same goes for figure S4e.
11. Figure 5a: I would strongly suggest to add an orthogonal section that shows that the bacteris are actually inside the CNS and not just attached to it.
12. Figure 5c,e: The legends do not fit the graphs!
13. Figure 6a: “count in brain”: It should be made clear that this is not the bacterial count in the parenchyma, but includes those bacteria that are found inside the capillaries.
14. Drosophila strains: There are 2 RNAi-lines against lpr2 listed. Which one has been used? It is important to state this either in general (if just one has been used) or per experiment if different RNAi-lines have been used. I guess “UAS-lpr-RNAi” is supposed to read “UAS-lpr1-RNAi”
15. In vivo larval infection: It is important to describe here who the larvae have been kept after injections – which temperature, which food...?
16. TEM: What is 1X PHEM buffer? Is this supposed to mean 0.1M PHEM buffer?
17. Bacterial quantification in infected Drosophila brains: what does an optimized number of slices mean? Was this the same slice distance for every brain? Is it below the bacterial diameter? Please clarify.

18. Figure S3a: What is the genotype *cyl+*? I think it is not mentioned anywhere.
19. Figure S4c, S6c: scale bar missing.
20. Figure S5b: "CFU counts in the brain": This is not true. This includes bacterial counts in the brain as well as attached to the brain. Please make this clear. The same goes for S5c.

Reviewer #3:

Remarks to the Author:

Review:

An original model of brain infection identifies the hijacking of host lipoprotein import as a bacterial strategy for blood-brain barrier crossing.

The study introduces an innovative screening system to identify microbes able to cross the Blood Brain Barrier (BBB) in *Drosophila*. To demonstrate the value of such a pioneering screening platform, the authors selected one of their identified microbes (GBS) and documented the sequence of events the bacteria initialize to cross the BBB. In addition to already known bacterial strategies to breach biological barriers, such as proteolytic activity or acidification, the authors identified the bacterial lipoprotein-like protein Blr responsible to organize transport across SPG cells. Blr binds the LDL receptor LpR2, which is also a target of *drosophila* lipoproteins, and initiates cellular receptor-mediated absorption of the bacteria. Although it remains unclear how the microbes exit into the brain, the authors successfully demonstrate that the molecular route is conserved and GBS enter the vertebrate brain using a similar strategy. The experimental design and data representation is throughout, and the established assay and scientific findings are of great scientific interest. I strongly recommend the manuscript for publication and have just minor comments/suggestions (see below).

1. Assay design

1.1 The authors cultivate the larval explants at 30°C, a temperature beyond the reproductive range of *Drosophila* and conditions known to trigger a heat stress response. Lower temperatures (between 20 and 25°) are certainly more suitable and reduce the proliferation rate of the bacteria. I do not ask for any experimental re-design/reproduction, however, a more detailed explanation in the manuscript will help scientist in the field.

1.2 Comment: The authors decided not to use a feeding approach in their experimental design. I recommend including such approach into their future studies and comparing the invasive routes across both barriers. Especially the cellular exit of microbes could help to enlighten the last stages of transcytosis.

2. Results

2.1 The larval explants expose fat body, wing disc and other tissues to bacteria. Did these peripheric tissues endocytose bacteria with similar rates? Moreover, after bacterial incubation – do absorbed bacteria leave the invaded cells again? PNGs are polarized cells, the apical site facing the periphery. It would be a valuable (but not essentially required in this manuscript) information to know if polarity plays a role for the transcytosis of bacteria.

2.2 Comment: The fact that Lamp1 positive compartments are loaded with microbes comes somehow with a little surprise to me (maybe a GOF artefact due to the overexpression of Lamp1-GFP?) – i would imagine to have a split route in Rab7 late endosomes, into not yet matured to Lamp1 positive compartments (similar to lysosome related organelles) and canonical lysosomes. Maybe a future idea is to use the published endogenous Rab-library to track down the migration route of invasive bacteria in different cell types.

2.3 The LrP2 data are sound, however, some bacteria still manage to cross the BBB in LrP2 loss of function experiments. Like published, lipoprotein particle receptors show some degree of redundancy and have a multitude of ligands. It remains unclear to me if the bacterial Blr lipoprotein represents a specific ligand to LrP2 or if the mere expression profile of LrP2 and other LDL receptors in SPGs is

responsible for the successful microbial transport across the BBB. If LrP2 is specific for Blr protein than the knowledge about its binding domain could have implications for drug-development.

2.4 *Drosophila* brains are densely packed. I wonder if all invasive bacteria remain close to the BBB or are able to migrate further into the brain. Since the authors speculate that the proteolytic activity of GBS is responsible to breach the lamellar layer of the BBB – a migration within the CNS may point into the same direction. Maybe it is possible to review existing imaging data and quantify the position of microbes in relation to the BBB?

2.5 Microbes were found after 3h in mice injected with GBS and the study reports bacteria associated with vascular endothelium cells (which form the BBB in most vertebrates). This is interesting since in a close vascular system high shear forces normally prevent undirected attachment to cellular surfaces. In addition, it was shown that the streaming velocity in the open circulatory system of *Drosophila* is very high implicating similar conditions. If invasive bacteria withstand high shear forces and adhere with mice endothelium cells/*drosophila* BBB glia than the authors may include another step into the sequence of events: the microbial adherence to cellular surfaces (last figure/discussion).

NCOMMS-20-14177

RESPONSE TO REVIEWERS' COMMENTS

First of all, we would like to thank very much the reviewers for their shared support of our work. We were hoping to offer to the community a novel model for investigating the mechanisms and consequences of brain infection, and we are very happy to see that it has been perceived as such.

For ease of revision, all major additions in the text have been highlighted in yellow in the revised manuscript. The text was also remodeled to fit the journal limits.

Report from Reviewer #1:

This manuscript is a fascinating new model to examine how bacteria penetrate barriers using fly larvae. Fly genetics allows for more rapid screening and the authors demonstrate in detail the methods to visualize and interpret analyses in such a small organ. The cell biology is strong and explained well. It is highly likely that many labs will try the model and hopefully the field will be advanced by this new experimentation.

Comments:

1) Could the authors add in an estimate of how the fly larval stage compares with the mouse developmental day (embryonic day ?)? It is telling that only GBS, a neonatal pathogen, is strong in the model and this may relate to the developmental age of the system; please comment.

Drosophila is a holometabolous insect with four life stages: embryonic, larval, pupal and adult stages. Comparing the fly larval stage with precise days of mouse developmental is as such complicated. Nevertheless, we choose the larval stage for the combination of an active neurogenesis with a functioning BBB, which is formed and sealed at the end of embryogenesis in *Drosophila*. Although late embryonic stage would have also been a possibility, being in a close system would have made our investigation very challenging.

To our delight, our model indeed revealed a link between mechanisms of infection by specific pathogens and life stage, as we found that LpR2 expression (through *LpR2::GFP* knock-in) at SPG barrier level is not detectable in adult flies (Fig. 4d compared to Fig. 4c). This reinforces our use of the larval stage both in *ex vivo* and *in vivo* set ups and might account for the particular success of GBS in crossing the BBB in this model. The corresponding text now reads:

Lines 150-152 of Results: *“Interestingly, we were not able to detect LpR2::GFP in the SPG of adult CNS (Fig. 4d), a striking result underlying the existence of different possible mechanisms depending on the life stage.”*

2) The discussion could comment on the lack of any previously found determinant to be active in their model. It is highly unlikely that all other studies are wrong. What are the limitations of the model?

Using the *ex vivo* set up, we have not found any major role for GBS capsule, hemolysin or cell wall anchored proteins using the sortase A mutant. However, one should remember that in the sortase A mutant, cell wall proteins are still synthesized and can associate with the plasma membrane. We have not performed a systematic screening of previously known GBS adhesins/invasins (HvgA, Srr1/2, SfbA, FbsC, PI2a pilus) which are sometimes strain-specific. As we wanted to take advantage of our model to offer a different perspective on the identification of GBS factors essential for brain entry, we choose to focus on previously untested factors. We thus cannot rule out that some of these factors could also be active in *Drosophila*.

This set up has indeed some technical limitations: i) the use of a lower temperature (30°C) than the one encountered in mammalian hosts (37°C), which can affect pathogen growth and fitness, as well as expression of temperature-dependent virulence factors; and ii) the lack of circulating immune cells, what could explain for example the results obtained with *Cryptococcus neoformans*. In addition, some mammalian factors targeted by GBS could be missing in *Drosophila*. These issues have now been incorporated as follows:

Lines 10-14 of Discussion: *“It is worth noting that fly experiments were performed at 30°C, and not at 37°C, the usual environment of mammalian pathogens, to allow Drosophila*

*development. This constitutes a limitation of our model since expression of some virulence factors can be temperature dependent.
Using our model, we aimed to identify novel factors crucial for BBB crossing by GBS...”*

3) Figure 5 panels C and E: the inset legends do not match the lines in the graphs. This is currently uninterpretable.

We are sorry for this mistake which has been fixed in the revised version of the manuscript (Fig. 5c and 5e).

Report from Reviewer #2:

Benmimoun et al. use *Drosophila* as a model organism to screen for factors implicated in host-pathogen interactions. They succeed in identifying the route that streptococcus agalactiae uses to pass the blood brain barrier in *Drosophila*. They combine ex vivo as well as in vivo infection setups to analyze molecular interactions between bacterial and fly proteins. Further, they show that the route of entry used in the fly is likely to be conserved in mammals. Overall, I find this study very interesting and recommend its publication. However, I have some concerns that I think need to be considered before publishing the study.

Major concerns:

1. I have some doubts about the disruption of the ECM and the PG cell layer. In Figure 2c The authors show GBS infected brains stained for GBS and *vkg::GFP*. They claim that these stainings show a disruption of the ECM by GBS. The top view indicates holes in the ECM where the GBS are (is this a single plain?). However, the orthogonal view rather indicates slight bending of the ECM. In this view I cannot see a strong difference between the non-infected and the infected ECM. Thus, I have strong doubts about the destruction of the ECM here.

We thank the Reviewer 2 for raising this point. We are sorry if the top view of original Fig. 2c was misleading. As stated by the reviewer, it is due to the bending of this layer, likely coming from the additional volume of the bacteria pressed onto the coverslip. We actually do not think to be able to visualise a complete destruction of the ECM/Col IV, at least in this short timeframe. With respect to what is known for other organs, it is likely that the ECM layer is deposited constantly, and *vkg* is at least highly expressed in the adult BBB (DeSalvo et al., 2014). As such a localised destruction of this layer might be transient and difficult to catch. We however detected a number of abnormalities in *Vkg::GFP* staining under GBS infection, including localised loss and bacteria sandwiched between multiple *Vkg* layers, which is what we tried to convey with the following sentence: “Whereas a collagen IV layer was still present under GBS infection, it appeared disturbed, generally weaker and sometimes absent where the bacteria were detected.” To better illustrate and support our conclusions, we have now changed the pictures of Fig. 2c, added additional pictures as Fig. S2a and also determined the contribution of acidosis to these changes. In addition, we have added a staining for Perlecan (*trol*) showing similar patterns (Supp. Fig. 4b-c). We have also updated Fig. 7 accordingly.

The corresponding text now reads:

Lines 36-41 of Results: “*First, using protein trap lines (*vkg::GFP*³⁷ and *Trol::GFP*³⁸) to visualise conserved components of the ECM (respectively collagen IV and heparan sulfate proteoglycan (HSPG) Perlecan), we revealed: i) that GBS was laying on or embedded in the ECM (Fig. 2c) and ii) that both the overall collagen IV and Perlecan networks were disrupted (Supp. Fig. 2a-b) and appeared locally clumped around the embedded bacteria (Fig. 2c and Supp. Fig. 2c). In addition we observed a decrease in signal intensity of *Trol::GFP* (Supp. Fig. 2b-c).”*

Lines 61-64 of Results: “*Blocking medium acidification using a HEPES buffer rescued Collagen IV general pattern (Supp. Fig. 2a), however localised clumping still remained around GBS (Fig. 2c). These results suggest that GBS could locally alter the ECM to facilitate access to the cellular layers.”*

Lines 73 of Results: “*including ECM rearrangement”*

Of note, we did try to generate a mutant of two genes encoding a potential collagenase in GBS. However, it kept reverting to wild-type, suggesting that such activity, or at least this protein, is essential to GBS fitness.

Further, concerning the damage of the PG layer: Figure 2e shows the expression of a membrane-bound GFP specifically in the PGs of infected, non-infected and infected+HEPES animals. The GFP signals appears much weaker in the infected brains vs. the non-infected brains or the infected, HEPES-treated brains. The authors claim that this change in GFP signal is due to a destruction of the PGs by acidification of the milieu caused by the GBS. Depending on how these pictures were taken, however, I have strong doubts about this interpretation. The GFP usually used in transgenic flies is highly pH-sensitive. It hardly fluoresces at a pH of 5 (Roberts et al. Scientific Reports 2018). Since the pH that was measured in the medium during infection is 4.6 (!) a change in the GFP signal might not be due to destruction of the PGs and thus loss of membrane-bound GFP, but simply to loss of GFP fluorescence, at least if those pictures have been taken live. Since I do not know how fast GFP-fluorescence recovers after exposure to low pH, I suggest that the same experiments should be redone with fixed and anti-GFP stained samples to make sure, that there is really a loss of GFP protein.

As stated by the reviewer, the GFP protein, and thus its fluorescence, is highly sensitive to pH. We were well aware of this caveat. In accordance, all the experiments relying on the interpretation of a GFP signal (PG>CD8-GFP as well as LpR2 MiMIC; Vkg::GFP, Trol::GFP, Lachesin::GFP; Rab5-GFP; FYVE-GFP; Lamp1-GFP) had already been performed in fixed conditions with an anti-GFP staining. We can thus interpret the decrease in the GFP signal used to mark the PG layer as a consequence of PG destruction during GBS infection (in acidic conditions, without HEPES) rather than a pH-induced loss of fluorescence. We are sorry if this experimental point was not better stated in our original version. We have now included the following statement in the Methods (*Drosophila* immunocytochemistry):

“Of note, due to medium acidification upon infection, all GFP fusions were detected with an anti-GFP antibody.”

2. Effect of acidification on the BBB. The authors claim that the addition of HEPES reverses the destructive effect on the BBB. The data using a permeability assay seem valid, but I have problems with the EM pictures. In figure 2g the authors show an EM picture of wt infected with GBS + HEPES and claim that the phenotype is reversed. Since there is no picture of wt + GBS without HEPES, I have no idea what the phenotype on EM-level would be. I would strongly suggest adding an EM-picture of wt+GBS. In Figure S2d there is a picture of a non-infected wt, but it has a different magnification. Further there is a zoom of the same picture as in 2g in S2e. I do not know what additional information this should give. I suggest showing EM-pictures of wt non-infected, wt infected and wt infected + HEPES all at the same magnification in Figure 2.

We have now grouped TEM pictures (non-infected, infected by WT GBS with or without HEPES) in Supp. Fig. 2e of the revised version, and moved the confocal pictures to main figures (2e and 3e). We hope that drawing similar conclusions from these two approaches makes the point convincing.

3. I further have a general doubt about the importance of this acidification for in vivo infections. The medium (Schneider's medium) used in the ex vivo infection setup has not much to do with the hemolymph that is filling the body cavity in vivo. Probably also the bacterial load is not similar to in vivo infections. Can the hemolymph in vivo really be that

strongly acidified without killing the animal immediately? Is this a realistic scenario or is the idea rather a local acidification there? Maybe it would be a good idea to measure hemolymph pH of infected vs. non-infected animals (in vivo model). In addition: Are there signs of PG/SPG destruction in the in vivo infection model?

We agree with the reviewer that *ex vivo* and *in vivo* sets up have intrinsically different parameters. As such these two approaches complement each other, and shine light on different aspects of the infection mechanisms that could be overridden or hidden in one of them.

In the case of medium acidification, GBS is known to secrete lactic acid, implying that the pH is locally lowered around the bacteria. In our explant system, in which bacteria proliferate freely, we propose it leads to the acidification of the whole medium, resulting in a systemic effect that allows us to reveal this phenomenon. However, we do think that similar events happen *in vivo*, albeit back at a more local scale. We indeed did not see any pH change in the hemolymph of larvae infected by WT GBS, as assayed by larva bleeding on pH paper. However, we do see signs of SPG destruction in the *in vivo* set up, as we initially mentioned in the discussion (lines 65-66 of the Discussion). It actually correlates with the amounts of bacteria found inside the brain. We have now added a **picture of SPG destruction *in vivo* as Supp. Fig. 5a**, and modified the text as follows:

Lines 189-190 of Results: *“We were also able to observe an altered SPG layer in brains with high bacterial counts (Supp. Fig. 5a).”*

Further, one could infect *ex vivo* brains using bacterial mutants that are not able to acidify the medium anymore (unfortunately I am not a specialist on GBS and do not know by which mechanism this happens, and thus cannot suggest something specific).

That would be a great experiment that we also considered. However, previous studies invalidating the GBS lactate dehydrogenase (*ldh*) gene in other Streptococci species found that it led to compensatory mutations, as well as alteration of bacterial growth (for example: *S. pyogenes*: Oehmcke-Hecht et al., Front Microbiol, 2017; *S. pneumoniae*: Gaspar et al., Infect Immun, 2014). We thus thought it would be difficult to interpret the resulting phenotype as only coming from loss of acidosis, and that pH buffering is actually a more direct evidence for it.

4. Putative biofilm formation by GBS delta *blr*. The authors claim that figure S3d shows putative biofilm formation in delta *blr* infected brains. This is not obvious. What is meant by “additional large structures”? Could you please mark them somehow? In addition, is this also visible in 3e? I would expect that it should be. If this is the case, why are there additional pictures in S3? In addition, as for the EM experiments before, the controls are missing. How does the infected wt look without HEPES? How does delta *blr* infected look without HEPES? How does figure Sf2 relate to this – I think this figure is not even mentioned in the text. I would recommend presenting the EM pictures in one figure together with the proper controls.

We understand the reviewer’s point and will try to clarify our observations. Our aim was to illustrate that GBS can be found on the surface of the larval brain as individual cocci, in chain and clusters embedded or not in a biofilm-type matrix. The extracellular matrix forming the biofilm often composed of carbohydrates is secreted by individual bacteria, and thus was present in Fig. S3e, as expected by the reviewer, albeit in more discrete, localised aggregates on the bacteria themselves rather than in the shape of a matrix embedding the bacteria. Formation of a proper biofilm implies a large concentration of bacteria, and is rarer.

We have now grouped the SEM pictures of cocci for the different strains and conditions in Fig. 3d, and coloured what we call biofilm matrix in yellow. We still decided to keep the picture of a proper biofilm structure, also colorised, for *Ablr* as Supp. Fig. 3d, as it is a rarer event. We also noticed while doing so that there has been mislabeling of one condition previously (*Ablr* HEPES versus non HEPES), and this has been corrected.

To support our initial SEM observation that GBS can form biofilms on the surface of the *Drosophila* larval brain, we searched for a lectin able to recognize the biofilm matrix and found that Concanavalin A is able to stain GBS matrix. These new data are shown in Supp. Fig. 3e. The corresponding text has been changed as follows:

Lines 97-100 of Results: *“Moreover, using SEM, we observed no obvious morphological difference between WT GBS and Ablr, that we found attached to the brain surface and in chains, and displaying biofilm-type matrix on their surface (Fig. 3d, matrix colorised in yellow).”*

Lines 111-117 of Results: *“In addition, using SEM, we did not notice detectable difference in the morphology of Ablr mutants with or without medium acidification (Fig. 3d). Interestingly, we also detected on Ablr-infected brains large film-like structures embedding bacteria and reminiscent of the polysaccharidic coat produced during biofilm formation (Supp. Fig. 3d, colorised in yellow). Using a marker for polysaccharides (the lectin Concanavalin A, see Methods), we confirmed that both WT GBS and Ablr mutants were actually able to form biofilms on the Drosophila larval brain (Supp. Fig. 3e).”*

Supp. Fig. 2f, originally mentioned Line 72 of the Results, was used to show that buffering medium acidity did not prevent GBS to form large clusters on the surface of the brain, suggesting that lower bacterial count within the brain does not strictly come from an upstream problem in adhesion. We have now removed this picture to avoid confusion.

5. LPR2 localization. Figure 4b and figure S4b-c do not agree on the localization of Lpr2. Figure 4b shows staining of an LPR2-GFP protein trap, while S4b-c shows stainings using an antibody against Lpr2 in wt animals(?). The LPR2-GFP protein trap seems to be much more abundant and more evenly distributed along the membrane than suggested by the anti-LPR2 staining. Further, the localization of the anti-LPR2 stained LPR2 at the SPG membrane is not obvious in figure S4b-c. I do not see colocalization with the SPG-membrane marker. I rather get the impression that the staining is found in the PGs. Here, I think better pictures would be important.

Since fusing a GFP to a protein might interfere with its localization, function or stability, it is important to clarify which staining reflects the “wild-typic” situation. As a first step I would suggest costaining *lpr2*-GFP animals for GFP and LPR2 to see if the stainings overlap. If this is the case one must assume that the GFP-fusion is more stable/localizes differently than the wt protein and cannot be used to draw any conclusions.

We fully agree with the reviewer that the localisation of Lpr2::GFP and of endogenous Lpr2, as detected by the anti-Lpr2 antibody, appeared different, even though they both marked the BBB. However, we believe this is due to the difficulty and unreliability of the anti-Lpr2 staining rather than an aberrant localisation of Lpr2::GFP. The anti-Lpr2 antibody we use (Parra-Peralbo and Culi, 2011) indeed works very poorly, as reported by J. Culi who generously shared this reagent and advice on immunostaining procedures. Accordingly, performing anti-Lpr2 staining on Lpr2::GFP very rarely produces an overlap, and when so, only in restricted zones of the brain surface. After trying additional protocols, we have managed to get a better staining of the endogenous Lpr2, even though it is still highly variable. We have now added the corresponding pictures (Supp. Fig. 4e-f) and updated the staining protocol.

To address the adequacy of Lpr2::GFP as a proxy for Lpr2, we have now:

i) shown that LpR2::GFP expression matches (Supp. Fig. 4b) what had been published previously in other tissues using anti-LpR2 antibody (Parra-Peralbo and Culi, 2011).

ii) infected homozygous and heterozygous flies with GBS and demonstrated that GBS entry is not different than for a control, “wild-typic” background (w¹¹¹⁸). This demonstrates that, at least on the role of LpR2 in brain infection by GBS, the parameter we address in this study, LpR2::GFP behaves similarly to endogenous LpR2 (Supp. Fig. 4c).

iii) shown that *LpR2* knockdown in the SPG does lead to a decrease in LpR2::GFP staining in the brain surface (Supp. Fig. 4d).

iv) Observed co-localisation between LpR2::GFP and GBS in membrane vesicles from the SPG, showing that LpR2::GFP can be endocytosed like wild-type LpR2 (Fig. 4g).

Altogether, we believe these new data address the issue of the use of LpR2::GFP as a proxy for LpR2 expression (and role) in the SPG during GBS infection.

The corresponding text now reads:

Lines 144-150 of Results: *“To assess LpR2 expression in the larval brain, we used an endogenous LpR2-GFP fusion (LpR2::GFP MiMIC line) resulting from a gene knock-in. This line recapitulated previously known profiles of expression in other tissues (wing disc and egg chambers⁵³, Supp. Fig. 4b) and behaved in a wild-type fashion for GBS entry (Supp. Fig. 4c). We observed that LpR2::GFP strongly colocalised with a membrane marker for the SPG (Fig. 4c), a pattern lost upon RNAi-mediated knockdown of LpR2 in the SPG only (Supp. Fig. 4d). A similar expression in the SPG was detected using an anti-LpR2 antibody (Supp. Fig. 4e-f).”*

Methods, *Drosophila* immunocytochemistry: *“Anti-LpR2 staining is highly variable, and permeabilisation in PBS-Triton 1% was used to help penetration.”*

6. In vivo infection model: I have some doubts about the in vivo infection model. Why do 60% of the mock-injected wt larvae die within 4 hours? Such a high mortality rate that is not associated with an infection, seem to be a bad basis for evaluating effects of an infection. Further, I do not understand, why the survivals have not been scored until all (at least the infected) animals are dead? Do any of the mock-injected wt animals survive? At what temperature are these experiments performed? If it is 30°C as for the ex vivo infection, this might be part of why they die, since animals usually are rather stressed at 30°C. Have infections using adult animals been tried? I understand that all other *Drosophila* data comes from larvae and thus a larval infection model makes sense, but if it is so difficult to infect them without killing them independent of the infection, it might make sense to switch to the adult here. Adults survive injections very well.

Larval injection is indeed tricky, much more than in the adult as stated by the reviewer. However, our goal was to develop a model in which the impact of brain infection on neurogenesis could ultimately be assessed. Despite the challenge, it was then necessary to validate our approach *in vivo* in the larval stage, during which extensive and well-characterised neurogenesis happens, contrary to the adult.

In addition to this overarching consideration, it was also crucial to stay in the same stage to validate our findings on BBB crossing, as we could not exclude that different entry mechanisms could take place at different developmental/life stage. In this line, we were not able to detect LpR2 expression in the adult SPG. These data have been added to further support our model of infection in a developing brain (Fig. 4d and see also response to Reviewer 1, point 1).

Lines 150-152 of Results: *“Interestingly, we were not able to detect LpR2::GFP in the SPG of adult CNS (Fig. 4d), a striking result underlying the existence of different possible mechanisms depending on the life stage.”*

Regarding the other experimental points raised by the reviewer:

1) 60% of the mock-injected larvae die due the injection itself. We believe it is mostly due to the difficulty of sealing the punctured cuticle, a scarring which is actually witnessed by the formation of a melanised spot.

2) We originally scored survival up to 4 h post-injection for the following reasons: i) all GBS-injected animals die between 4 h and 5 h post-injection (for an inoculum $8,8 \times 10^8$ CFU / ml of hemolymph) and ii) mock-injected larvae can survive very late, up to adulthood. We thus wanted to stop the experiment at a timepoint for which all conditions could be censored in the same way, and with all the animals at the same developmental stage to avoid taking into consideration other parameters. We discussed this approach with the Computational biology department of the Institut Pasteur. The biostatistician we interacted with did not see any major issue with this approach. Although tracking is stopped before animal death, censoring information is fully part of our datasets, meaning that the fact that death happens later is integrated in the analysis. Consequently, we have kept our original dataset, however with revised statistics using an adjusted p-value calculated through Holm-Sidak (see below), which led to the same conclusions. Nevertheless, we have added to the revised version of the manuscript an additional survival curves of mock-injected and GBS-wt-injected larvae until all infected animals were dead (Supp. Fig. 5b) to explain the choice of our timing.

We also added the following explanation:

Methods, *In Vivo* Drosophila larval infection: “*As all animals injected with WT GBS died between 4 h and 5 h post-injection while mock-injected larvae could survive up to adulthood, thus passing several developmental stages, we decided to score our conditions until 4 h post-injection to avoid further variable parameters.*”

3) These experiments were performed at 30°C as a compromise between what Drosophila could withstand and a temperature that would reach the closest to the 37°C at which GBS normally infects host cells. We agree that it could, at least partially, explain the high lethality we observed even with the mock. However, similarly to the point raised above, we thought it was important to keep as many parameters constant between *ex vivo* and *in vivo*. These different points are now addressed as follows:

Methods, *In vivo* Drosophila larval infection:

“*Mock injection results in lethality per se, due to a combination of experimental limits:*

- i) unsuccessful healing of the punctured cuticle, which should be sealed by a melanisation spot as witnessed in surviving larvae;*
- ii) potential damages to tissues neighbouring the injection point, which are favoured by muscular contraction of the larva during injection;*
- iii) potential temperature-induced stress (30°C).”*

Also, the text states that the mutant bacteria are less virulent than wt bacteria. Having a look at the survival curves, I would say that they are avirulent, since there is no difference between non-infected animals and those infected with mutant bacteria. This also fits the mouse data.

Reviewer 2 is right, and we have now specified the avirulence of $\Delta lgt/lsp$ and Δblr mutants in the revised version of the manuscript.

Legend of Fig. 5c: “*c. Kaplan-Meier survival curves for larvae injected with mock, WT GBS, $\Delta lgt/lsp$, Δblr and $\Delta blr+blr$ strains ($n = 60$ for each condition) show that $\Delta lgt/lsp$ and Δblr are avirulent”*

Concerning the *lpr2*-RNAi survivals: How has the statistical analysis been done? Were the entire curves compared to each other or just the single time points? I can hardly believe that there should not be a significant difference between wt+GBS and SPG>*Lpr2*-RNAi GBS (graph says n.s., even though the text states there is a difference - please clarify). On the other hand, it is difficult to believe that there is really a difference between non-infected and infected SPG>*Lpr2*-RNAi animals. Has a multiple comparison using Holm-Sidak been done? I would strongly suggest talking to an expert on statistical analysis of lifespan data. Also: the legends do not fit the graphs (figure 5c,e)!

The survival curves were compared to each other using the Log-Rank (Mantel-Cox) test. After discussing the analysis with the biostatistician from the Institut Pasteur, he agreed on this strategy, while suggesting to calculate an adjusted p-value to account for pairwise comparisons between several conditions. We compared a stack of p-values, using Holm-Sidak as proposed by the reviewer. In this case, there is still a non-significant difference between Control + GBS and SPG>*LpR2 RNAi* + GBS (the p-value changes from 0.057 to 0.16, shifting even more into non-significance). However the difference between SPG>*LpR2 RNAi* Non-infected and SPG>*LpR2 RNAi* + GBS loses its significance, with an adjusted of-value of 0.13 (compared to 0.034 originally). This would fit into our proposition (lines 206-207 of Results in our initial manuscript) that lethality might not come primarily from brain infection, but a consequence of systemic infection and high bacteremia.

The figures 5c and 5e have been modified to take into account this new analysis and the legends include the new adjusted p-values. The text now reads:

Lines 210-213 of Results: *“Survival curves showed that depleting LpR2 in the SPG did not significantly alter lethality compared to wild-type animals (compare black and orange curves in Fig. 5e). This suggests that, although lethality might result from brain infection, it mainly depends on a systemic effect and the infection of other organs and compartments.”*

Bacterial counts in the brain are shown for 4h p.i. and 18h p.i. (or 16h as stated in the figure? – please clarify). How many animals are even alive at 18h post infection? Are these then just those animals that have not been properly injected? I have severe doubts that this number tells anything about the infection process.

Bacterial counts shown at 16 h post-infection were performed using a lower inoculum 10^5 GBS/ml of hemolymph (compared to 8.8×10^8 CFU/ml of hemolymph), as stated in the legend of the Figure 5a. These larvae did survive longer, however we did not score lethality precisely throughout the experiment. Our goal here was to show that we could find conditions in which we generated brain infection with a longer survival time that would allow analysis of defects in neurogenesis.

To avoid confusion, these counts were removed from the revised version of the manuscript.

If I understand the data correctly, Figure 5 b, d and S5a show bacterial counts within the brain based on brain stainings and manual counting of bacteria that are inside the brain. In contrast figure S5b,c are based on bacterial counts as determined in homogenized tissue by plating the homogenate and counting colonies. Thus “brain” in S5b,c includes all bacteria within the brain and attached to the brain. This is an important difference that should be made clear.

The reviewer is right. Figure S5b,c show bacterial counts determined in homogenised infected tissue after washing. Thus, these counts include bacteria attached to the brain in addition to

those inside, as we stated in line 195 of Results in our initial manuscript. This difference has now been precised in the legends of Supp. Fig. S5c-d of the revised version of the manuscript.

The number of bacteria in the brain after 4h of infection is very low. I have difficulties to believe that such a low number of bacteria would kill the animal. This should be discussed.

As stated above, we believe a main component, if not the major cause, of lethality, is primarily systemic infection. We have now strengthened this point by altering our initial sentence (lines 206-207 of Results in our initial manuscript) as follows:

Lines 212-213 of Results: *“This suggests that, although lethality might result from brain infection, it mainly depends on a systemic effect and the infection of other organs and compartments.”*

7. Mouse infection model: The authors claim that there are “destroyed blood capillaries” found in GBS infected mouse brains. They refer to Figure s6a. In S6a one might believe this, however in S6b, which I think shows the same mouse-bacterium combination, this does not seem the case. Are there really destroyed capillaries, or is this due to sectioning? Maybe a 3D reconstruction of the capillaries could help here, compared to non-infected animals of course. If the authors want to show a difference between the capillaries, control stainings of non-infected animals should anyways be shown.

Very well spotted by the reviewer. Supp. Figure 6b was mistakenly marked as “4 h post inoculation”. It actually shows a sagittal brain section from a naive (non-infected age-matched control) mouse, reflecting also the difference between the intact capillaries (S6b) and the destroyed capillaries after infection (S6a). The figure has been corrected along with the corresponding legend, in the revised manuscript.

Minor concerns:

1. Acknowledgments: mdr65-mtdt-tomato à mdr65-mtd-tomato

This error has been fixed in the revised version of the manuscript.

2. Figure 1a: I do not understand why the hemolymph is depicted as spheres instead of filling up the whole animal? Also, the authors talk about “brain cell populations”, but just show neurons and neglect cortex glia, astrocyte-like glia and ensheathing glia.

We did not want to mention all glial subtypes, that we are well aware of, for the sake of simplicity in comparing with the mammalian brain. We have now removed the spheres from the drawing of the larva in the revised version of the manuscript and precised that neurons are only one type of brain cell populations in the corresponding figure legends.

3. Figure 1d. scale bars are missing. In the orthogonal sections I guess the more actin-rich region is the neuropil. Where is it in the left-most picture. Are the stacks comparable? We have now added the scale bars. The stacks are comparable, they show the entire thickness of the VNC. The actin-rich region is indeed the neuropil. It does not display the same appearance upon GBS infection, as we know there are neuronal disorganisation in this case (another aspect of infection that we will address in further studies).

4. Figure 2a: Which region is shown? What is the dashed line? Figure 2a (“Attached”) represents the outer layer (extremity) of the brain with dashed line

representing the brain limit with the SPG in red. This explanation has been added to the legend of the revised version of the manuscript.

5. Figure 2c: the figure legend says vkg::GFP, the figure says COIIV::GFP. I would suggest sticking to the fly nomenclature here.

We changed this and kept the fly nomenclature in the revised version of the manuscript.

6. Figure 2e: the non-infected +HEPES control is missing

This control has been added.

7. Figure 3: A and A' should be separated in the legend

We separated a and a' in the legend of the revised version of the manuscript.

8. Figure 4a: Please add gene abbreviations in the legend (shg, arr, mgl)

We added gene abbreviations in the legend of Figure 4a in the revised version of the manuscript.

9. Figure 4c: why is there no anti-BLr blot of the brain lysates (since they are also an “input”) to show that they are negative? Similar in Figure S4d: Why are the bacterial inputs not probed? They should be negative for Lpr2-GFP.

We have now added the corresponding blots in Supp. Fig. 4g-h.

10. Figure 4 e-f: what are the schematic representations on the left? They are not mentioned anywhere (even not in the legend). Please either explain them or take them out. Further, the merge is not very helpful the way it is presented since the “white” channel is too prominent. What is the blue channel in the merges? It is not mentioned anywhere. The same goes for figure S4e.

The blue channel represented the DAPI. To avoid any confusion and redundancy, the schematic representations and the merge pictures have been removed from the revised version of the manuscript.

11. Figure 5a: I would strongly suggest to add an orthogonal section that shows that the bacteris are actually inside the CNS and not just attached to it.

We have added the corresponding orthogonal section in the Fig. 5a of the revised version of the manuscript.

12. Figure 5c,e: The legends do not fit the graphs!

This error has been fixed in the revised version of the manuscript.

13. Figure 6a: “count in brain”: It should be made clear that this is not the bacterial count in the parenchyma, but includes those bacteria that are found inside the capillaries.

The corresponding figure legend (6a) has now been changed to include the requested clarification.

14. Drosophila strains: There are 2 RNAi-lines against lpr2 listed. Which one has been used? It is important to state this either in general (if just one has been used) or per experiment if different RNAi-lines have been used. I guess “UAS-lpr-RNAi” is supposed to read “UAS-lpr1-RNAi”

We now have left the unique RNAi line against *LpR2* we used for the figures in the Methods. The error about *UAS-LpRI RNAi* has been fixed in the revised version of the manuscript.

15. In vivo larval infection: It is important to describe here who the larvae have been kept after injections – which temperature, which food...?

We described better the experimental setting of the injected larval survival in the revised version of the manuscript.

Methods, In vivo *Drosophila* larval infection: *“The injected larvae were kept on standard fly food plates placed in a 30°C incubator with 60% humidity during scoring.”*

16. TEM: What is 1X PHEM buffer? Is this supposed to mean 0.1M PHEM buffer? It is indeed 0.1M PHEM buffer. This mistake has been fixed in the revised version.

17. Bacterial quantification in infected *Drosophila* brains: what does an optimized number of slices mean? Was this the same slice distance for every brain? Is it below the bacterial diameter? Please clarify.

An optimized number of slices takes in consideration the numerical aperture of the objectives as well as the laser used. In our case, as the slides were imaged in the same conditions, it was 0.38 μm between each slice. This is below GBS diameter (0.5 to 2 μm). We clarified this point in the Methods section (*Drosophila* immunocytochemistry) of the revised version of the manuscript, which now reads:

“Samples were mounted in Mowiol mounting medium and visualised with a laser scanning confocal microscope (Zeiss LSM 880), with an optimal distance between each slice of 0.38 μm .”

18. Figure S3a: What is the genotype *cyl+*? I think it is not mentioned anywhere. *GBS-cyl+* is a hyper-hemolytic mutant of *GBS NEM316* secreting more hemolysin, as was stated in the Results section (line 80, now).

19. Figure S4c, S6c: scale bar missing. Missing scale bars have been added.

20. Figure S5b: “CFU counts in the brain”: This is not true. This includes bacterial counts in the brain as well as attached to the brain. Please make this clear. The same goes for S5c. The requested clarification has been added in the revised version of the manuscript (legends Supp. Fig. 5c-d).

Reviewer #3 (Remarks to the Author):

Review:

An original model of brain infection identifies the hijacking of host lipoprotein import as a bacterial strategy for blood-brain barrier crossing.

The study introduces an innovative screening system to identify microbes able to cross the Blood Brain Barrier (BBB) in *Drosophila*. To demonstrate the value of such a pioneering screening platform, the authors selected one of their identified microbes (GBS) and documented the sequence of events the bacteria initialize to cross the BBB. In addition to already known bacterial strategies to breach biological barriers, such as proteolytic activity or acidification, the authors identified the bacterial lipoprotein-like protein Blr responsible to organize transport across SPG cells. Blr binds the LDL receptor LpR2, which is also a target of *drosophila* lipoproteins, and initiates cellular receptor-mediated absorption of the bacteria. Although it remains unclear how the microbes exit into the brain, the authors successfully demonstrate that the molecular route is conserved and GBS enter the vertebrate brain using a similar strategy. The experimental design and data representation is throughout, and the established assay and scientific findings are of great scientific interest. I strongly recommend the manuscript for publication and have just minor comments/suggestions (see below).

1. Assay design

1.1 The authors cultivate the larval explants at 30°C, a temperature beyond the reproductive range of *Drosophila* and conditions known to trigger a heat stress response. Lower temperatures (between 20 and 25°) are certainly more suitable and reduce the proliferation rate of the bacteria. I do not ask for any experimental re-design/reproduction, however, a more detailed explanation in the manuscript will help scientist in the field.

We initially chose this temperature as a general compromise between larval viability and aiming for a temperature as close as 37°C as possible. We agree that it would be good to see how much we gain or lose on each side depending on the temperature. In addition, it would be crucial to tweak these conditions for each pathogen, to find the best conditions for their entry. These considerations are now explained throughout the manuscript:

Lines 9-11 of Results: *“The culture medium was then inoculated with the chosen pathogens at selected doses at 30°C, close to mammalian body temperature yet tolerated by Drosophila.”*

Methods, Culture of *Drosophila* brain explants: *“30°C was chosen as a compromise temperature allowing Drosophila development (although with some potential heat response compared to the more standard 25°C) while culturing mammalian pathogens closer to their usual environment (37°C).”*

1.2 Comment: The authors decided not to use a feeding approach in their experimental design. I recommend including such approach into their future studies and comparing the invasive routes across both barriers. Especially the cellular exit of microbes could help to enlighten the last stages of transcytosis.

We agree with Reviewer 3 that a feeding assay for the *in vivo* part would enlighten us on how GBS manages to cross different types of barrier tissues, especially the gut. In our case, we favoured direct injection as a first approach to be able to control precisely the bacterial dose that would be loaded into the circulation, and to avoid the variability linked to feeding rates

and, indeed, passage of the gut epithelia. We were feeling more confident to address the point of BBB crossing for GBS in these conditions. For future studies it would be very interesting to add this approach especially for neurotropic pathogens known to be transmitted through contaminated food such as *Listeria monocytogenes*.

2. Results

2.1 The larval explants expose fat body, wing disc and other tissues to bacteria. Did these peripheral tissues endocytose bacteria with similar rates? Moreover, after bacterial incubation – do absorbed bacteria leave the invaded cells again? PNGs are polarized cells, the apical site facing the periphery. It would be a valuable (but not essentially required in this manuscript) information to know if polarity plays a role for the transcytosis of bacteria.

We did not look in details into bacterial entry in other peripheral tissues, while noticing bacteria attached to various tissues, including discs and the lymph gland.

Although we did not analyse in a live-imaging set-up whether (and how) absorbed GBS leave invaded cells, we believe it must since: i) we observed bacteria outside the BBB, either close or deeper (neuropile), what shows that GBS travels through different layers of cells within the brain; and ii) ultimately, GBS is an extracellular bacteria, and we indeed did find it localised between cells.

The SPG is polarised along an apico-basal axis, and in further studies it would be interesting to assess its relevance to GBS translocation, although it might be tricky to play with it due to the crucial role of the BBB as a neuroprotective filter.

2.2 Comment: The fact that Lamp1 positive compartments are loaded with microbes comes somehow with a little surprise to me (maybe a GOF artefact due to the overexpression of Lamp1-GFP?) – i would imagine to have a split route in Rab7 late endosomes, into not yet matured to Lamp1 positive compartments (similar to lysosome related organelles) and canonical lysosomes. Maybe a future idea is to use the published endogenous Rab-library to track down the migration route of invasive bacteria in different cell types.

GBS withstands low pH and actually thrives in acidic environment, such as the vagina it colonises with a pH of 4. In the case of cellular translocation, it has actually been shown that GBS is found and survive in macrophages phagolysosomes (for a review see Shabayek and Spellenberg, 2017), which exhibit a very low pH (around 4.5) and contain various toxic factors for bacteria. GBS seems to turn on a specific acid response to live in these compartments, and phagosomal acidification is actually essential for the intracellular survival of GBS, a usually extracellular bacteria. Our data are in line with these findings, and it is thus possible that GBS ends up in and uses lysosomes to translocate through the SPG. Even though we cannot rule out an artefact from Lamp1-GFP overexpression, we also find similar results with the overexpression of *spinster* (Spin, Supp. Fig. S4k), known to label late endosomal/lysosomal compartments.

We have now added a picture showing LpR2::GFP colocalising with GBS in SPG membrane vesicles (Fig. 4h), and added a the following sentence:

Lines 22-23 of Discussion: “...*the exact endocytic journey of the GBS(Blr)-LpR2 complex still remains to be precisely demonstrated.*”

In future studies, it would be interesting to use the extensive endogenous Rab-library to precisely track the possible routes for bacterial translocation depending on the cell type.

2.3 The LrP2 data are sound, however, some bacteria still manage to cross the BBB in LrP2 loss of function experiments. Like published, lipoprotein particle receptors show some degree of redundancy and have a multitude of ligands. It remains unclear to me if the bacterial Blr lipoprotein represents a specific ligand to LrP2 or if the mere expression profile of LrP2 and other LDL receptors in SPGs is responsible for the successful microbial transport across the BBB. If LrP2 is specific for Blr protein than the knowledge about its binding domain could have implications for drug-development.

A first possibility could be that the RNAi knockdown is not complete and as such there is some residual LpR2. We have now added confocal pictures of *LpR2 RNAi* in the SPG showing the extent of knockdown (Supp. Fig. 4d). Another option would be that GBS could use other routes, independently of lipoprotein receptors. Finally, we agree with Reviewer 3 that lipoprotein receptors show some degree of redundancy. Here in particular, LpR1 and LpR2 are very similar and we were surprised to find that *LpR1* knockdown in the SPG did not affect GBS entry did. This might be simply explained by a difference in the expression profile of these two receptors, as proposed by Reviewer 3. Transcriptional profiling of the BBB suggests that *LpR1* is not expressed in the SPG (a mistake in our previous version and which has now been corrected). However, the knockdown of *mgl*, which is expressed in the SPG (Brankatschk et al, 2014), does not affect GBS entry (Fig. 4a). This suggests that at least Blr has preferred binding partners. Ultimately, a refined analysis of the expression profile of the different lipoprotein receptors in the brain will shed more light on the specificity of this interaction.

We have now added the following sentence in the text:

Lines 141-142 of Results: *“Interestingly, knocking down the closely related and partially redundant LpR1⁵³ did not decrease GBS entry.”*

We fully agree that understanding the protein domains involved in Blr and LpR2 interactions, from both sides, would help drawing pharmacological strategies. In this line, we bet that the LRR domain of Blr was the best candidate. We generated a GBS mutant in which the LRR region was precisely deleted (*blr^{ΔLRR}*). To our surprise, the *blr^{ΔLRR}* mutant was still able to bind to LpR2::GFP. We have now incorporated this new set of data as Supp. Fig. 4i-j. They are described in the text as follows:

Lines 165-170 of Results: *“We next asked whether the LRR domain of Blr was essential for such interaction. We generated a GBS mutant in which the LRR region was deleted (*blr^{ΔLRR}*, see Methods and Supp. Fig. 4i). Using the same co-immunoprecipitation strategy, we found that LRR-deleted Blr was still able to bind to LpR2::GFP, ruling out a strict requirement of LRR region for this interaction (Supp. Fig. 4j). Altogether these data demonstrated that Blr is able to bind LpR2 in a LRR-independent manner.”*

Lines 29-31 of Discussion: *“This interaction does not seem to require the LRR domain of Blr (Supp. Fig. 4j), a surprising result entailing that histidine-triad domain of Blr should be considered as potential interactor and interesting pharmacological target.”*

2.4 Drosophila brains are densely packed. I wonder if all invasive bacteria remain close to the BBB or are able to migrate further into the brain. Since the authors speculate that the proteolytic activity of GBS is responsible to breach the lamellar layer of the BBB – a migration within the CNS may point into the same direction. Maybe it is possible to review existing imaging data and quantify the position of microbes in relation to the BBB?

That is a great point and something we wanted to do was to plot GBS position with respect to the SPG for different genetic or infection conditions. However, it was quite tricky and,

ultimately, we were not sure it would be as informative as we would have hoped due to the SPG structure. Indeed, the SPG/BBB could be described as a rugby ball pierced by repeated tunnels. This makes all cells within the brain at a maximal distance of around 50 μm from the BBB, possibly to ensure all cells have a reasonable access to systemic factors transmitted through the BBB. In the case of GBS translocation, the fact that we find cocci at different distances from a BBB signal indicate they did travel. However, the BBB structure means that how much one specific bacterium has migrated throughout the brain cannot be inferred/assumed from its position, since we have no record of its face of entry.

2.5 Microbes were found after 3h in mice injected with GBS and the study reports bacteria associated with vascular endothelium cells (which form the BBB in most vertebrates). This is interesting since in a close vascular system high shear forces normally prevent undirected attachment to cellular surfaces. In addition, it was shown that the streaming velocity in the open circulatory system of *Drosophila* is very high implicating similar conditions. If invasive bacteria withstand high shear forces and adhere with mice endothelium cells/*drosophila* BBB glia then the authors may include another step into the sequence of events: the microbial adherence to cellular surfaces (last figure/discussion).

Indeed, microbial adherence to the BBB is a prerequisite before invasion and actually most studies on the mechanisms behind brain entry by GBS identified factors involved in this step (such as the pilus). We thank the reviewer for bringing this issue to our attention. We now mentioned this first step in our summary model (Figure 7) and in the text:

Lines 53-58 of Discussion: *“How GBS adheres to the Drosophila brain is a crucial step that remains to be determined in our model. The ECM is a layer rich in glycosaminoglycans recognised by many pathogens⁶⁶, and the fly ECM indeed contains HSPGs, including Perlecan (Supp. Fig. 2b-c). Moreover several HSPGs, such as the PG-secreted Dally-like⁶⁷, were shown to be important for GBS adhesion to Drosophila S2 cells⁶⁸ as well as for virulence using an infection model in which adult flies were pricked with GBS serotype Ia (A909 strain)⁶⁹.”*

Reviewers' Comments:

Reviewer #2:

Remarks to the Author:

Benmimoun et al. have now dealt with most reviewer concerns. This has further improved the manuscript. I thus, recommend it for publication now. The only concerns I still have, and I think they should still be addressed are the following: The control (non-infected) staining is missing in S5a. The fact that no acidification of the hemolymph has been found in the in vivo infection model should be included in the text and discussed. This suggests that if acidification plays a role in vivo, it is more likely to occur locally.

Reviewer #3:

Remarks to the Author:

the authors responded to all my points sufficiently and i recommend the manuscript for publication.

NCOMMS-20-14177A

RESPONSE TO REVIEWERS' COMMENTS

Report from Reviewer #2:

Benmimoun et al. have now dealt with most reviewer concerns. This has further improved the manuscript. I thus, recommend it for publication now. The only concerns I still have, and I think they should still be addressed are the following: The control (non-infected) staining is missing in S5a. The fact that no acidification of the hemolymph has been found in the *in vivo* infection model should be included in the text and discussed. This suggests that if acidification plays a role *in vivo*, it is more likely to occur locally.

We have added the corresponding BBB staining of a VNC from a mock-injected larva 4 h after injection (Supp. Fig. 5a).

Regarding the acidification of the hemolymph, we are reluctant here to state more regarding this issue. We indeed assessed hemolymph pH from a few GBS-injected larvae through bleeding on pH paper, as stated in our previous response. However, as BBB destruction is a relatively rare event *in vivo* (2 out of 13 brains, 15% as shown in the Methods), we cannot exclude that we missed these larvae. In addition, other events could happen *in vivo*, which would need to be carefully addressed and which are beyond the scope of this paper. In consequence, we have added the following sentence in the text to stay on the cautious side while addressing the request for discussion of the Reviewer:

Discussion Lines 44-47: “This suggests that acidosis-linked alterations of the BBB might be a conserved mechanism taking place during GBS infection, likely localised around concentrations of bacteria releasing lactic acid. In addition, other events could also account for BBB destruction *in vivo*.”